# In-Context Learning can Perform Continual Learning Like Humans

## Abstract

Large language models (LLMs) can adapt to new tasks via in-context learning (ICL) without parameter updates, making them powerful learning engines for fast adaptation. While extensive research has examined ICL as a few-shot learner, whether it can achieve long-term retention and cross-task knowledge accumulation when multitasks arrive sequentially remains underexplored.h Motivated by human memory studies, we investigate the retention characteristics of ICL in multitask settings and extend it to in-context continual learning (ICCL), where continual learning ability emerges through task scheduling and prompt rearrangement. Experiments on Markov-Chain and regbench based benchmarks demonstrate that, for specific large-language models, ICCL benefits from distributed practice (DP) in a manner analogous to humans, consistently revealing a spacing "sweet spot" for retention. Beyond retention performance, we propose a human-retention similarity metric to quantify how closely a continual-learning (CL) method aligns with human retention dynamics. Using this metric, we show that linear-attention models such as MAMBA and RWKV exhibit particularly human-like retention patterns, despite their retention performance lagging behind that of Transformer-based LLMs. Overall, our results establish ICCL as cognitively plausible and practically effective, providing an inference-only CL paradigm that enables LLMs to maintain long-term retention of a target task under cross-task interference without parameter updates.

## 1 Introduction

Large language models (LLMs) have demonstrated remarkable capabilities across a wide range of domains, with in-context learning (ICL) emerging as one of their most distinctive properties (Brown et al., 2020). Via ICL, LLMs adapt new tasks on the fly by conditioning on demonstrations embedded in the prompt without parameter update. This inference-only property turns them into versatile, efficient solvers that cope with zero-shot (Ouyang et al., 2022), few-shot (Brown et al., 2020), and many-shot (Agarwal et al., 2024) regimes, spanning Bayesian inference (Xie et al., 2024), regression (Raventós et al., 2023; Wu et al., 2023), and reinforcement learning (Laskin et al., 2023; Lee et al., 2023; Wang et al., 2025a). Such flexibility suggests that ICL could evolve into a general-purpose learning engine rather than remain a mere few-shot adapter (Wang et al., 2025b; Fan & Liu, 2025). However, prior work predominantly focuses on single-task or one-off adaptation behavior. Only limited studies investigate multi-task ICL settings, and these works typically optimize short-term prediction by mixing or reordering demonstrations rather than examining whether knowledge can persist across tasks. Thus, it remains unclear whether ICL can support continual learning, where knowledge must be maintained and reused across sequentially presented tasks.

In the broader literature, catastrophic forgetting (CF) has long been recognized as a fundamental challenge in gradient-based continual learning (GBCL). To mitigate CF, a rich body of GBCL has been developed, including regularization-based approaches that reshape the objective function (Li et al., 2020; Evilevitch & Ingram, 2021), replay-based approaches that reorganize the data stream (Rolnick et al., 2019; Buzzega et al., 2021), and approaches that directly modify the optimization process (Arous et al., 2021; Poggio et al., 2011). Although effective on many benchmarks, GBCL approaches fundamentally depend on parameter modification or explicit memory buffers and often struggle to achieve a desirable balance between stability and plasticity. This naturally raises a broader question: *does a similar form of forgetting also arise in parameter-free, inference-only regime of LLMs?*

Table 1: Property comparisons among GBCL, ICL and ICCL.

| Property | GBCL | ICL | ICCL |
|---|---|---|---|
| Parameter updates | Yes | No | No |
| Update rules | Gradient Descent | Blackbox Computation | Blackbox Computation |
| Knowledge carrier | Parameters | Memory/Hidden States | Memory/Hidden States |
| Task settings | Multiple Tasks | Single Task | Multiple Tasks |

In other words, when tasks are presented sequentially within a long context, can a LLM stably retain the mapping of a target task while processing intervening tasks? To investigate this question, we introduce **in-context continual learning (ICCL)** as a framework for studying inference-time retention in LLMs. Unlike multi-task ICL, which jointly learns all tasks, ICCL focuses on how the ordering and distributed scheduling of target-task and intervening-task demonstrations affect the model's ability to preserve the target-task behavior throughout extended inference, without any parameter updates.

Several recent studies have begun to explore directions related to ICCL, often by extending ICL with prompt modification (e.g., prompt tuning) or external augmentation (e.g., memory retrieval) (Wang et al., 2022; Gao et al., 2023; Momeni et al., 2024; Shinwari & Usama, 2025). While these approaches show promise, they also present several limitations. First, most efforts remain confined to single-task adaptation, without systematically addressing multitask sequential adaptation and cross-task knowledge accumulation. Second, they often rely on heuristic prompt engineering or auxiliary components that raise complexity yet remain vulnerable to memory bloat and retrieval errors, and plateau when procedural memory integration rather than fragmented retrieval is required. Third, current evaluation benchmarks are often either too simple, resulting in trivial zero-shot solutions, or too complex, exceeding the capacity of LLMs to learn within context. As a result, they fail to adequately support systematic analyses of multitask learning dynamics.

In this paper we focus on ICCL, which exploits the LLM's built-in memory to flexibly control both learning and retention rather than external plug-and-play modules and manually specified learning rules, as summarized in Table 1. To compare GBCL and ICCL and to expose ICCL's characteristics, we empirically examine the following perspectives: First, in his pioneering studies of human memory, Hermann Ebbinghaus deliberately used "nonsense syllables" in serial-learning tasks (Ebbinghaus, 2013) to strip away prior knowledge and semantic associations, thereby isolating the core mechanisms of memory. Guided by the same principle, we design two benchmarks for evaluating LLMs whose parameters already encode vast human knowledge: randomly generated Markov chains and regular language generated by random finite automata. These chains preserve essential dynamical-system properties (Attal, 2010; Froyland, 2001) while minimizing contamination from pre-existing knowledge. Second, cognitive science shows that structured practice schedules enhance long-term memory of human (Cepeda et al., 2006; 2008; Pavlik Jr & Anderson, 2005; Bothell, 2020), underscoring the need to introduce task scheduling in machine continual learning. Guided by this, we inject explicit task identifiers and cognition-inspired schedules to systematically probe how scheduling and prompts shape the retention characteristics of both LLMs and GBCL. Those includes single practice (SP), multiple practice (MP), and distributed practice (DP). Experiments show that ICCL retains information more robustly than GBCL baselines and consistently privileges DP over SP and MP, mirroring cognitive findings on human memory. Moreover, linear-attention models such as Mamba and RWKV exhibit retention dynamics that more closely parallel human behavior, even though their absolute retention lags behind standard self-attention. Our contributions are threefold:

- ICCL provides, to our knowledge, the first framework that systematically investigates whether forgetting arises purely at inference time for LLMs and how to enable them to maintain long-term retention of a target task under cross-task interference without parameter updates.
- We show that ICCL benefits from DP in a manner analogous to humans, consistently exhibiting a spacing sweet spot that enhances retention and provides a more favorable balance between stability and plasticity compared to GBCL baselines.

---

†Corresponding to:

- We introduce evaluation benchmarks that combine retention performance with human-retention similarity, enabling systematic analysis of multitask retention and revealing that linear-attention models (e.g., MAMBA, RWKV) display particularly human-like retention patterns, despite lower absolute performance than Transformer-based LLMs.

## 2 METHODOLOGY

### 2.1 FORMALIZATION OF ICCL

We now formalize the general protocol of ICCL. A task is denoted by $\tau : y_i^\tau \sim p_\tau(\cdot|x_i)$, which specifies a conditional distribution mapping an input $x_i$ to an output $y_i^\tau$. A segment of $\varphi$ experiences on task $\tau$ is represented as $\mathcal{D}_\varphi^\tau = \{x_1^\tau, y_1^\tau, \ldots, x_\varphi^\tau, y_\varphi^\tau\}$. To model ICCL, we consider a historical sequence of such experience segments $\mathcal{C}_t = \bigoplus_{i=1}^N \mathcal{D}_{\varphi_i}^{\tau_i}$, where $t = \sum_{i=1}^N \varphi_i$ and $\oplus$ denotes ordered concatenation. Unlike a random collection, the order of segments is essential since ICCL is sensitive to the task order arrangement. Given a query input $x$ and the historical sequence $\mathcal{C}_t$, ICCL aims to approximate the output of an unknown target task $\tau^*$. Formally, we define:

$$\hat{p}_\theta(y|x, \mathcal{C}_t) = \mathcal{F}_\theta\left(\bigoplus_{i=1}^N \mathcal{D}_{\varphi_i}^{\tau_i}, x\right) \Rightarrow p_{\tau^*}(y|x), \tag{1}$$

where $\mathcal{F}_\theta$ denotes a sequential prediction structure parameterized by $\theta$, such as a large language model. Here, $p(y|x, \mathcal{C}_t)$ represents the probability of predicting output $y$ given $x$ and $\mathcal{C}_t$, and the goal is to approximate the true conditional distribution $p_{\tau^*}(y|x)$. Intuitively, ICCL leverages prior task experiences in context to infer the correct behavior on the target task. Finally, tasks $\tau_i$ and $\tau^*$ are elements of a task collection $\mathcal{T}$, which defines the general scope of tasks considered in ICCL.

**Induction of task identifier.** Previous zero-shot instruction-following studies typically assume the availability of a task description $d^\tau$ that supplements the prompt (Tanaka et al., 2024; Lou & Yin, 2023). However, in practice, such meta-information is rarely known a priori and tasks must often be inferred implicitly. To accommodate this more realistic setting, we relax $d^\tau$ into a *task identifier*, which need not encode semantic details but merely serves to distinguish one task from another. This abstraction aligns well with real-world applications: for instance, in preference modeling, the system may not have access to a user's full profile but can instead rely on a user ID combined with the interaction history. Under this formulation, ICCL can be reformulated as

$$\hat{p}_\theta(y|x, \mathcal{C}_t) = \mathcal{F}_\theta\left(\bigoplus_{i=1}^N \left(d^{\tau_i}, \mathcal{D}_{\varphi_i}^{\tau_i}\right), d^{\tau^*}, x\right), \tag{2}$$

where task identifiers $d^{\tau_i}$ and $d^{\tau^*}$ act as lightweight labels that allow the model to keep track of task boundaries without requiring explicit semantic descriptions.

**Evaluation of retention.** Retention in ICCL refers to the model's ability to preserve and use earlier demonstrations within long contexts. Given a historical sequence $\mathcal{C}_t$, we next evaluate how much information about the target task $\tau^*$ is retained after the model processes an intervening block $\mathcal{D}_{\varphi_D}^{\tau'}$ from another task $\tau' \neq \tau^*$. Specifically, this evaluation quantifies the extent to which the model "remembers" the target task $\tau^*$ after the $t$-step sequence $\mathcal{C}_t$ and an additional $\varphi_D$-step exposure to the distracting task $\tau'$:

$$R_{\tau^*, \tau', \mathcal{C}_t, \theta}(t + \varphi_D) = \mathbb{E}_{x, \tau'}\left[M\left(\hat{p}_\theta(\cdot|x, \mathcal{C}_t \oplus D_{\varphi_D}^{\tau'})), \ p_{\tau^*}(\cdot|x)\right)\right], \tag{3}$$

where $M(p_1, p_2)$ is a metric that quantifies the similarity between two probability distributions $p_1$ and $p_2$. We use *Normalized Performance* (denoted by the symbol $R$) to assess the accuracy of $\hat{p}_\theta$, with $R(t) = 0$ indicating a complete loss of memory and $R(t) = 1$ indicating a perfect match between the estimator and the ground truth $p_{\tau^*}$.

**Scheduling strategy.** We formalize a scheduling strategy to capture how task demonstrations are arranged in a historical sequence $\mathcal{C}_t$. Specifically, different practice conditions correspond to different constructions of $\mathcal{C}_t$, as summarized as follows:

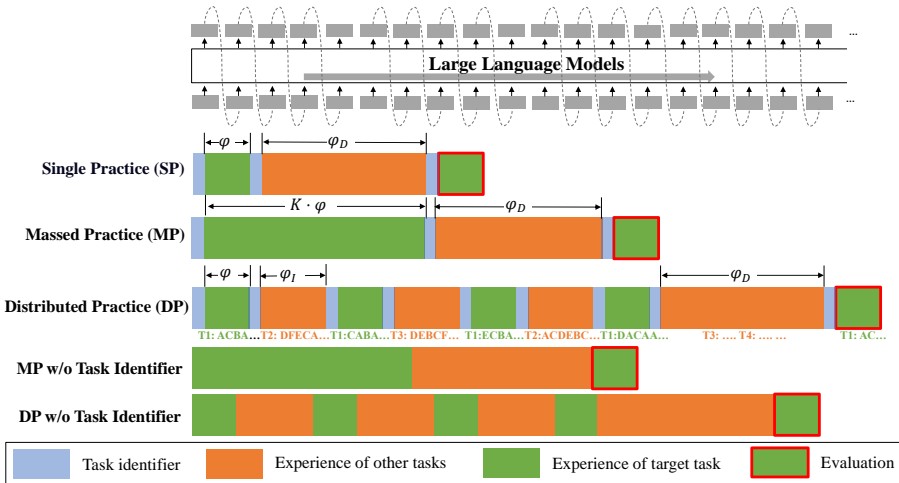

Figure 1: Illustration of how target and other task demonstrations are arranged under SP, MP, and DP, together with their variants without explicit task identifiers in a historical sequence. During the interleaved period, MP contains only target-task demonstrations (with $\varphi_I = 0$) while DP includes both target-task and intervening-task demonstrations ($\varphi_I > 0$). The post-exposure period ($\varphi_D$) always involves multi-task interference and is used to evaluate retention decay.

$$
\begin{aligned}
\text{Single Practice (SP):} \quad & \mathcal{C}_t = \mathcal{D}_\varphi^{\tau^*} \\
\text{Massed Practice (MP):} \quad & \mathcal{C}_t = \mathcal{D}_{K\varphi}^{\tau^*} \\
\text{Distributed Practice (DP):} \quad & \mathcal{C}_t = \bigoplus_{i=1}^{K} \big(\mathcal{D}_\varphi^{\tau^*}, \mathcal{D}_{\varphi_I}^{\tau_i}\big), \quad \forall \tau_i \in \mathcal{T}, \ \tau_i \neq \tau^*.
\end{aligned}
\tag{4}
$$

Eq. (4) characterizes three canonical scheduling conditions, which are illustrated in Figure 1. In *SP*, demonstrations of the target task $\tau^*$ are presented only once in a contiguous block of length $\varphi$, after which the sequence moves to other tasks. In *MP*, the demonstrations of the target task are extended to a larger block of length $K\varphi$, corresponding to the repeated rehearsal of $\tau^*$ without interruption. In *DP*, the sequence alternates between the target task and other tasks: each target task block of length $\varphi$ is followed by another task block $\mathcal{D}_{\varphi_I}^{\tau_i}$ with $\tau_i \neq \tau^*$. This setup more closely resembles human retention patterns (spacing is most effective when intervening content is distinct from the target) and allows ICCL to retain the target task knowledge in a manner analogous to human retention. Therefore, we maximize differences between target and other tasks to create clear contextual boundaries and reliable reactivation cues. Compared with independent ICL, which must repeatedly include large blocks of target-task demonstrations to sustain retention, ICCL achieves comparable performance using far fewer demonstrations. Under DP, the target-task pattern is naturally reinstated through spaced exposure, enabling the model to maintain high retention even when only a limited number of target-task demonstrations are available. This advantage becomes essential when retention must persist over long content lengths.

To further investigate the role of explicit task identifiers, we additionally evaluate two variants where $d^\tau$ is removed from the prompt, denoted as *MP w/o instruction* and *DP w/o instruction*, respectively. These two variants test whether the presence of explicit task identifiers contributes to multitask sequential adaptation beyond the task order arrangement.

## 2.2 RETENTION DYNAMICS IN ICCL

A well-documented phenomenon in human retention is the spacing effect (Anderson et al., 2004; Cepeda et al., 2008): MP triggers rapid decay, whereas DP yields superior long-term retention. To account for this effect, the ACT-R cognitive model (Borst & Anderson, 2017; Pavlik Jr & Anderson, 2005; Petrov, 2006) describes the resulting learning and retention curves. Suppose the learner practices (observes) the target task $\tau^*$ at the time stamp of $t_1, t_2, \ldots, t_{K\varphi}$ and is exposed to other tasks otherwise; the ACT-R model predicts the probability of retaining the task at any subsequent

time $t = t_{K\varphi} + \varphi_D$ by the following equation:

$$\hat{R}(t) = 1/[1 + \exp(-\frac{w(t) - \gamma}{s})], \quad w(t) = \ln \sum_{i=1}^{K\varphi} [\kappa \cdot (t - t_i)]^{-d} \quad (5)$$

where $w(t)$ represents the memory activation that decays over time when no new practice of the target task occurs, and $\hat{R}_{\tau^*}(t)$ denotes the probability of successfully accomplishing task $\tau^*$ at time $t$ given the memory activation $w(t)$. Notice that in our settings, for MP, $t_i = i$, for DP, $t_i = i + \left\lfloor \frac{i}{\varphi} \right\rfloor \varphi_I$. The parameters $s$, $\gamma$, and $d$ are hyper-parameters to be determined. To bridge the gap between the actual time-dependency of human retention and the context length in ICCL, we introduce an additional hyper-parameter $\kappa$. Our experimental results in Section 3.4 reveal an interior optimum of $\hat{R}(t)$ under DP, indicating the existence of a spacing sweet spot that maximizes the retention probability. This empirical pattern mirrors decades of findings in human memory research and provides practical guidance for prompt scheduling that avoids both massed and overly spaced practice. This retention arises not from short-term contextual reuse, but from spaced reactivation of the target-task pattern, which counteracts interference and long-context attention decay.

### 2.3 QUANTIFYING HUMAN RETENTION SIMILARITY VIA MAHALANOBIS DISTANCE

We now ask a related question: *to what extent do LLMs resemble human memory in their retention characteristics?* To enable a direct human–model comparison, we introduce Human Retention Similarity via Mahalanobis Distance (HRS-MD), which evaluates how closely the fitted ACT-R parameters of a model fall within parameter distributions reported in cognitive psychology. Specifically, Let the fitted parameter vector be $\hat{\boldsymbol{\theta}} = [\hat{d}, \hat{s}, \hat{\gamma}]^\top$, where $\hat{d}$ denotes the decay rate, $\hat{s}$ the activation noise, and $\hat{\gamma}$ the retrieval threshold estimated from retention curves. Human reference distributions are characterized by a mean vector $\boldsymbol{\mu} = [\mu_d, \mu_s, \mu_\gamma]^\top$ and covariance matrix $\boldsymbol{\Sigma}$, derived from prior experimental studies (Pavlik Jr & Anderson, 2005; Lewis & Vasishth, 2005; Bothell, 2020; Borst & Anderson, 2017; Said et al., 2016; van der Velde et al., 2022). HRS-MD is then defined as $(\hat{\boldsymbol{\theta}} - \boldsymbol{\mu})^\top \boldsymbol{\Sigma}^{-1} (\hat{\boldsymbol{\theta}} - \boldsymbol{\mu})$, which quantifies the discrepancy between the model-fitted parameters and human baselines while accounting for parameter scales and correlations. A smaller HRS-MD indicates closer alignment between LLMs and human memory retention. This metric enables systematic comparisons across ICCL and GBCL methods, revealing which model classes most closely reproduce human-like retention pattern. Details of $\boldsymbol{\Sigma}$ and the ACT-R parameters, along with their reference distributions and the rationale for the Mahalanobis metric, are provided in Appendix A.2.

## 3 EXPERIMENT RESULTS

### 3.1 EXPERIMENT SETTINGS

**Models and Baselines.** We evaluate two categories of methods: inference-only ICCL models and GBCL baselines. The ICCL group consists of four representative open-source LLMs: LLaMA3-8B, DeepSeek-R1, MAMBA, and RWKV-7. Note that ICCL focuses on inference-time retention within standard LLMs and does not rely on architectural or external-memory mechanisms; thus, comparisons with memory-enhanced models fall outside the scope of this study. The GBCL group includes three classical approaches and one modern method: Stochastic Gradient Descent (SGD) (Arous et al., 2021), Experience Replay (ER) (Rolnick et al., 2019), Elastic Weight Consolidation (EWC) (Evilevitch & Ingram, 2021), and Prompt Gradient Projection (PGP) (Qiao et al., 2024). For ICCL, experiences serve as the context in LLMs without updating any parameters; for GBCL, we construct a small predictive model that maps the input & task-identifier to an output and sequentially refine it with the arriving experiences. The detailed model architecture and hyperparameters are provided in the Appendix A.3. To ensure fair comparisons, all models are evaluated on the same input sequences and prompt structures, with greedy decoding applied during inference. Each experiment is repeated 16 times with different random seeds, and results are reported as stable averages.

**Benchmark Tasks.** Discrete Markov Chains (DMC) are widely adopted benchmarking tasks (Zekri et al., 2024; Akyürek et al., 2024; Liu et al., 2024) because they capture essential properties of dynamical systems while minimizing contamination from pre-existing knowledge, thus providing

a clean and controllable environment for analyzing retention and interference dynamics. In our benchmark, two levels of complexity are considered: *simple* tasks with $N_s = 4$ states and *complex* tasks with $N_s = 8$ states. For each setting, transition matrices are generated to sample states and construct experience sequences. Each target and interference tasks are defined by a transition matrix $P_{target} \in \mathbb{R}^{N_s \times N_s}$ and a transition matrix $P_{interference} \in \mathbb{R}^{N_s \times N_s}$, and the model receives a state sequence as the main prompt content to predict the next state. Prompts are prepared under two conditions: (i) with task identifiers, where each sequence segment is prefixed with a label (e.g., `[TARGET_TASK]`, `[INTERFERENCE_TASK]`), and (ii) without identifiers, where the same content is presented unlabeled. This setting enables systematic evaluation of whether explicit identifiers facilitate multitask sequential adaptation. We apply the scheduling strategies introduced in Section 2.1 to form prompts in SP, MP, and DP formats. We include a detailed example in the Appendix A.4 illustrating how a DMC prompt is constructed, along with its corresponding transition matrices. We further design a complementary benchmark built upon the `RegBench` framework (Akyürek et al., 2024) to evaluate ICCL in natural language–style environments. `RegBench` introduces linguistically structured inputs and richer syntactic variation, which make `RegBench` closer to real-world sequential learning scenarios. Appendix A.4 provides complete details on the prompt construction process and ICCL's retention performance on this new benchmark.

**Parameter Settings.** We configure the DMC benchmark with $\varphi = 100, K = 5, \varphi_I \in \{10, 50, 100, 200, 400, 600\}, \varphi_D \in (0, 700)$, considering the token limits of different LLMs, e.g., `DEEPSEEK-R1` supports up to approximately 4096 tokens, whereas `LLAMA3-8B` allows up to 8192 tokens. These settings ensure that all models operate within their valid context windows while maintaining comparable information exposure across architectures. They also enable controlled studies on *interference length* ($\varphi_I$) and *duration* ($\varphi_D$), providing a consistent basis for analyzing retention and interference as the effective context grows. This unified configuration guarantees that each model runs at its optimal performance regime without exceeding its token budget.

**Evaluation Metrics.** A bottleneck in DMC evaluation is that performance divergence across tasks often stems from intrinsic task differences rather than prediction accuracy. To suppress this noise, we first introduce a normalized performance metric. Inspired by the normalization schemes in (Cha, 2007), we define the normalized retention performance metric $M$ (from eq. (3)) by rescaling the Bhattacharyya distance (Liu et al., 2021; Zekri et al., 2024) $D_{Bah}(p_1, p_2)$ to the interval $[0, 1]$:

$$M(\hat{p}_\theta, p_{\tau^*}) = \frac{\exp\big[D_{Bah}(p_{\tau^*}, p_{\tau^*}) - D_{Bah}(\hat{p}_\theta, p_{\tau^*})\big] - 1}{\exp\big[D_{Bah}(p_{\tau^*}, p_{\tau^*}) - D_{Bah}(p_{\mathcal{U}}, p_{\tau^*})\big] - 1}, \tag{6}$$

where $D_{Bah}$ is the Bhattacharyya distance and $p_{\mathcal{U}}$ denotes the uniform distribution, representing complete forgetting. This normalization makes retention scores comparable across tasks by rescaling the model–target divergence between the best case $D_{Bah}(p_{\tau^*}, p_{\tau^*})$ and the worst case $D_{Bah}(p_{\mathcal{U}}, p_{\tau^*})$. Bhattacharyya distance is chosen because it directly measures overlap between predictive and target distributions, though its raw scale varies with label entropy and task complexity. The exponential transform further linearizes distance differences and amplifies meaningful gaps when distributions are close, following standard similarity-normalization practices (Cha, 2007). Under this mapping, $M = 1$ corresponds to perfect retention, $M = 0$ to uniform guessing, and intermediate values reflect retention quality on a consistent cross-task scale. All reported results are given as mean $\pm$ 95% CI. We additionally employ HRS-MD to quantify alignment with human memory dynamics.

### 3.2 RETENTION PERFORMANCE UNDER SP, MP AND DP

We first evaluate how the retention performance of ICCL models and GBCL baselines varies on simple and complex tasks with task identifiers and $\varphi_I = 200$, under SP, MP, and DP scheduling. The experimental results on the complex task are shown in Figure 2. For both SP and MP, we observe that ICCL models exhibit a sharp performance drop immediately after the last target-task block, followed by a small continued decline as $\varphi_D$ increases, whereas some GBCL methods show a more gradual but consistent decrease–underscoring their susceptibility to catastrophic forgetting. Under DP, however, ICCL benefits substantially: DP effectively mitigates forgetting and maintains higher retention, while GBCL baselines show little improvement. We also observe that once LLM transitions from the target-task block to intervening tasks, its retention on the target task drops abruptly, even though the target demonstrations were encountered only moments earlier and remain relatively close

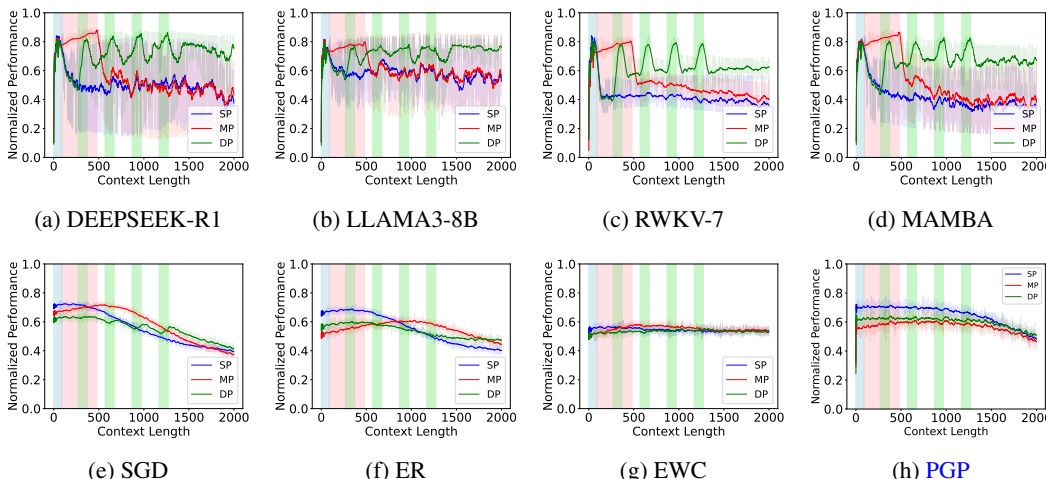

(a) DEEPSEEK-R1     (b) LLAMA3-8B     (c) RWKV-7     (d) MAMBA

(e) SGD     (f) ER     (g) EWC     (h) PGP

Figure 2: Retention performance on complex task under SP, MP, and DP settings (with task identifiers, $\varphi_I = 200$). Target task blocks are highlighted in light blue (SP), pink (MP), and green (DP).

Table 2: Retention performance (mean ± 95% CI) under DP for simple ($N_s = 4$) and complex ($N_s = 8$) Markov-chain tasks across varying intervals $\varphi_D$.

| | Simple Task (markov chain with $N_s = 4$) | | | | |
|---|---|---|---|---|---|
| Model Types | $\varphi_D = 0$ | $\varphi_D = 100$ | $\varphi_D = 200$ | $\varphi_D = 400$ | $\varphi_D = 600$ |
| LLAMA3-8B | $\mathbf{0.819 \pm 0.086}$ | $\mathbf{0.737 \pm 0.090}$ | $\mathbf{0.865 \pm 0.030}$ | $\mathbf{0.833 \pm 0.067}$ | $\mathbf{0.781 \pm 0.078}$ |
| MAMBA | $0.668 \pm 0.106$ | $0.655 \pm 0.105$ | $0.678 \pm 0.094$ | $0.685 \pm 0.100$ | $0.667 \pm 0.101$ |
| SGD | $0.606 \pm 0.100$ | $0.581 \pm 0.084$ | $0.558 \pm 0.086$ | $0.507 \pm 0.098$ | $0.471 \pm 0.101$ |
| ER | $0.608 \pm 0.095$ | $0.613 \pm 0.075$ | $0.617 \pm 0.077$ | $0.620 \pm 0.081$ | $0.629 \pm 0.085$ |
| EWC | $0.617 \pm 0.101$ | $0.612 \pm 0.086$ | $0.604 \pm 0.088$ | $0.601 \pm 0.099$ | $0.602 \pm 0.101$ |
| PGP | $0.665 \pm 0.064$ | $0.652 \pm 0.066$ | $0.637 \pm 0.059$ | $0.609 \pm 0.069$ | $0.562 \pm 0.079$ |
| | Complex Task (markov chain with $N_s = 8$) | | | | |
| Model Types | $\varphi_D = 0$ | $\varphi_D = 100$ | $\varphi_D = 200$ | $\varphi_D = 400$ | $\varphi_D = 600$ |
| LLAMA3-8B | $\mathbf{0.784 \pm 0.054}$ | $\mathbf{0.776 \pm 0.056}$ | $\mathbf{0.753 \pm 0.086}$ | $\mathbf{0.783 \pm 0.057}$ | $\mathbf{0.792 \pm 0.058}$ |
| MAMBA | $0.750 \pm 0.059$ | $0.700 \pm 0.061$ | $0.608 \pm 0.103$ | $0.724 \pm 0.064$ | $0.726 \pm 0.048$ |
| SGD | $0.577 \pm 0.072$ | $0.540 \pm 0.076$ | $0.491 \pm 0.080$ | $0.464 \pm 0.073$ | $0.448 \pm 0.085$ |
| ER | $0.519 \pm 0.081$ | $0.501 \pm 0.086$ | $0.472 \pm 0.088$ | $0.479 \pm 0.080$ | $0.495 \pm 0.091$ |
| EWC | $0.563 \pm 0.079$ | $0.551 \pm 0.087$ | $0.526 \pm 0.091$ | $0.537 \pm 0.083$ | $0.542 \pm 0.093$ |
| PGP | $0.608 \pm 0.057$ | $0.600 \pm 0.061$ | $0.584 \pm 0.060$ | $0.549 \pm 0.063$ | $0.515 \pm 0.060$ |

to the query in absolute context position. This sharp decline shows that the dominant failure mode is not insufficient recency, but rather the model's susceptibility to interference and long-range attention decay during continued context processing.

Building on the above results, Table 2 compares the retention performance of ICCL models and GBCL baselines under DP across simple and complex tasks as $\varphi_D$ increases from 0 to 600. ICCL consistently outperforms all GBCL methods, and although retention declines with larger $\varphi_D$, ICCL remains in a high-retention regime while GBCL methods operate at substantially lower levels. Although ER and EWC show relatively small numerical variations across $\varphi_D$, this pattern arises from early saturation at a low-retention regime. By contrast, ICCL maintains much higher retention with a slower degradation rate, demonstrating stronger resistance to extended interference.

**Ablation Study on Target Task Block Numbers $K$.** To further understand how the number of target-task blocks $K$ influences model retention, we conduct an ablation study where $K$ varies in $\{3, 5, 6, 7\}$ while fixing $\varphi_D = 600$. Figure 3 presents the normalized retention performance of ICCL and GBCL methods under DP. We observe that $K$ has a much stronger impact on ICCL methods than on GBCL methods. For transformer-based LLMs (e.g., DEEPSEEK-R1 and LLAMA3-8B),

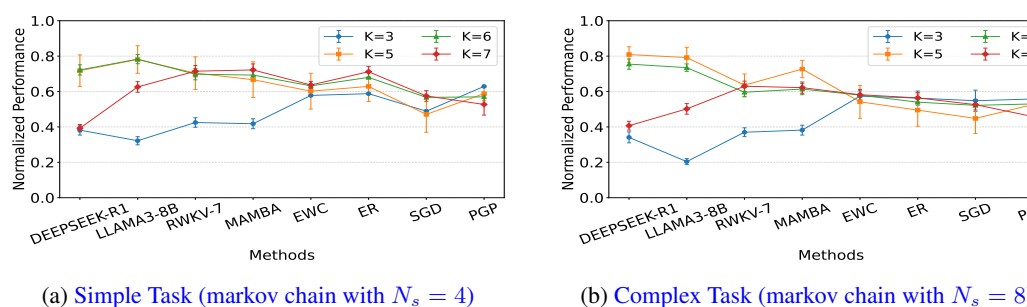

(a) Simple Task (markov chain with $N_s = 4$)  (b) Complex Task (markov chain with $N_s = 8$)

Figure 3: Retention performance (with $\varphi_D = 600$) under DP for simple ($N_s = 4$) and complex ($N_s = 8$) Markov-chain tasks across different target task block numbers ($K \in \{3, 5, 6, 7\}$).

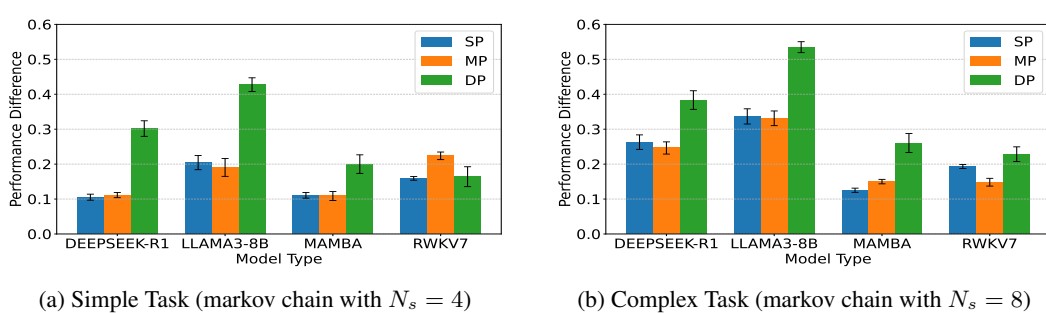

(a) Simple Task (markov chain with $N_s = 4$)  (b) Complex Task (markov chain with $N_s = 8$)

Figure 4: Retention performance difference of ICCL models with and without task identifiers under SP, MP, and DP ($\varphi_I = 200$ and $\varphi_D = 600$). Positive values indicate higher retention when task identifiers are included in the prompt.

retention performance first increases and then decreases as $K$ grows, because a larger $K$ yields longer effective contexts that approach or exceed the transformer's token limit. In contrast, linear-attention LLMs (e.g., RWKV-7 and MAMBA) show continuously improving and eventually stable retention performance, reflecting their superior scalability to long sequences. Overall, the results confirm that context length ($K\varphi$) interacts with model architecture to determine how efficiently previous task information is retained. This trend consistently appears in both simple and complex Markov-chain tasks, highlighting $K$ as a key factor shaping long-term memory dynamics in LLMs.

### 3.3 EFFECT OF TASK IDENTIFIER ON ICCL RETENTION

Figure 4 illustrates retention performance difference of ICCL models when prompts include task identifiers versus when they do not, under SP, MP, and DP conditions. Here, the plotted value represents the difference between ICCL's retention performance with and without identifiers; a positive value indicates an improvement when identifiers are used. As shown in Figure 4, incorporating task identifiers consistently enhances retention across all models and scheduling strategies, with the most pronounced gains observed under DP. The effect is more substantial in complex DMC tasks, suggesting that explicit identifiers become increasingly important as task complexity grows.

### 3.4 SPACING SWEET SPOT IN ICCL UNDER DP

Figure 5 reports the average retention performance of ICCL and GBCL models under DP across different intervals $\varphi_I$, evaluated on both simple and complex tasks. Here, average retention performance is defined as the mean retention over $\varphi_D$, thereby capturing memory retention beyond the immediate target-task exposure. The results reveal a clear contrast between ICCL and GBCL. For ICCL, DP induces a characteristic *spacing sweet spot*, where average retention performance peaks at an intermediate $\varphi_I$ lying between 100 and 400. This sweet spot consistently emerges across both simple and complex tasks, although average retention performance is higher in the simpler setting. These findings empirically support the presence of a spacing sweet spot and further demonstrate that

Table 3: Fitted ACT-R parameters with corresponding MSE and HRS-MD values.

| Model | Decay Rate ($d$) | Activation Noise ($s$) | Scaling Factor ($\kappa$) | Threshold ($\gamma$) | MSE | HRS-MD |
|---|---|---|---|---|---|---|
| LLAMA3 | 0.14 | 2.00 | 1.20 | 1.01 | 0.0024 | 500.22 |
| DEEPSEEK-R1 | 0.35 | 1.69 | 0.43 | -0.24 | 0.0155 | 302.39 |
| RWKV-7 | 0.27 | 1.62 | 0.94 | 0.59 | 0.0052 | **287.57** |
| MAMBA | 0.29 | 1.59 | 0.76 | 0.33 | 0.0049 | **272.74** |
| SGD | 0.41 | 2.00 | 0.77 | 0.15 | 0.0036 | 445.07 |
| ER | 0.27 | 2.00 | 1.36 | 1.02 | 0.0061 | 466.74 |
| EWC | 0.22 | 2.00 | 1.31 | 1.65 | 0.0021 | 481.53 |

ICCL under DP surpasses the performance achieved with SP and MP. Thus, ICCL benefits from DP in a manner analogous to human memory, and careful tuning of $\varphi_I$ can substantially enhance retention. In contrast, GBCL baselines exhibit no such sweet spot: their performance remains largely flat or gradually declines as $\varphi_I$ increases, indicating that DP provides little to no advantage.

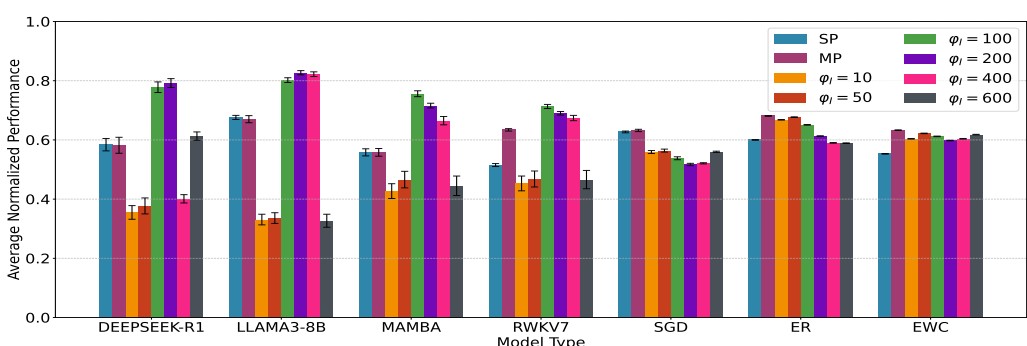

(a) Simple Task (markov chain with $N_s = 4$)

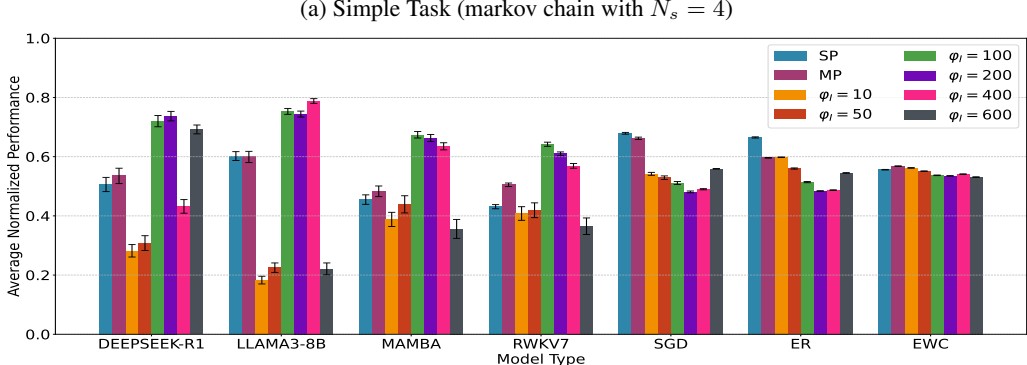

(b) Complex Task (markov chain with $N_s = 8$)

Figure 5: Average retention performance comparison among ICCL and GBCL methods under DP with different intervals ($\varphi_I$) on both simple and complex tasks.

### 3.5 HRS-MD QUANTIFICATION

To investigate how the aforementioned retention curves align with human retention, we employed the human-retention-validated ACT-R model to fit the ICCL and GBCL curves. Specifically, we used eq. (5) to model the normalized ICCL/GBCL performance obtained from eq. (3). Table 3 summarizes the fitted ACT-R parameters, mean squared error (MSE) during parameter fitting, and HRS-MD values for both ICCL and GBCL methods based on retention performance results on both simple and complex tasks under DP across different intervals $\varphi_I$. Several consistent patterns emerge. First, ICCL models–except for LLaMA3–achieve lower HRS-MD values than GBCL baselines, indicating

that their retention dynamics more closely approximate human memory. In particular, MAMBA and RWKV-7 yield the lowest HRS-MD scores ($272.74$ and $287.57$, respectively), demonstrating the strongest alignment with human retention patterns. By contrast, GBCL methods produce substantially higher HRS-MD values, reflecting greater divergence from human memory dynamics. Second, the fitted parameters reveal distinctive tendencies across the two paradigms. ICCL models generally converge to moderate decay rates, relatively low activation noise, and thresholds near zero—indicating efficient and accessible retrieval dynamics, particularly evident in MAMBA and RWKV-7. In contrast, GBCL methods exhibit more extreme parameter configurations: SGD yields the highest decay rate ($d = 0.41$), consistent with rapid forgetting, while ER and EWC converge to elevated thresholds ($\gamma \geq 1.0$), implying stricter retrieval conditions and reduced accessibility of stored knowledge. These patterns highlight the poor balance between stability and plasticity in GBCL approaches.

## 4 RELATED WORK

**GBCL** is the classical approach for mitigating catastrophic forgetting through parameter updates. Methods typically fall into three categories: EWC, ER, and SGD. EWC constrains updates to parameters important for previous tasks, using Fisher-information–based scores (Li et al., 2020; Evilevitch & Ingram, 2021) or online importance estimates such as synaptic intelligence (Zenke et al., 2017; Zenke et al.). ER maintains a small memory buffer and interleaves past examples with new data (Rolnick et al., 2019; Buzzega et al., 2021). SGD (Arous et al., 2021; Poggio et al., 2011) enables rapid adaptation but suffers severe forgetting due to its lack of preservation mechanisms. Recent extensions emphasize representation stability, but remain incompatible with our setting: EC-ACE (Xiong et al., 2024) relies on vision-specific multi-view consistency losses, while InsCL (Wang et al., 2024) adapts LLMs via instruction-tuned parameter updates—both outside the scope of inference-only ICCL.

**ICL** allows LLMs to adapt to tasks from in-prompt demonstrations without parameter updates. Prior work has analyzed its empirical sensitivities (Zhao et al., 2021; Liu et al., 2021; Lu et al., 2021; Liu et al., 2023) and offered theoretical views of ICL as Bayesian inference, gradient-based adaptation, or kernel regression (Xie et al., 2021; Akyürek et al., 2022; Von Oswald et al., 2023; Han et al., 2023). Many-shot studies further show extension to more complex tasks (Agarwal et al., 2024; Garg et al., 2022). Yet most research focuses on single-task adaptation, leaving sequential multitask retention largely unexplored. Recent continual-ICL efforts adjust prompts incrementally (Wang et al., 2022; Xiong et al., 2024) or use retrieval modules (Gao et al., 2023; Momeni et al., 2024; Shinwari & Usama, 2025), but these rely on heuristics or auxiliary components prone to memory growth and retrieval noise. While (Xiong et al., 2024) mixes demonstrations across tasks for short-term gains, ICCL instead examines long-term retention under sequential exposures.

**Human memory research** offers a rich foundation for understanding retention dynamics. Ebbinghaus (Ebbinghaus, 2013) introduced the forgetting curve, later refined into power-law formulations (White, 2001). The *spacing effect* shows that distributed practice yields superior long-term retention, a finding supported by large-scale meta-analyses (Cepeda et al., 2006; 2008). These phenomena are formalized in the base-level learning equation (Pavlik Jr & Anderson, 2005) and its ACT-R extension (Pavlik & Anderson, 2008; Bothell, 2020; Borst & Anderson, 2017), which link recall to the frequency and recency of exposures. Our work leverages these principles to test whether LLMs display analogous retention patterns when demonstrations are distributed across prompts.

## 5 CONCLUSION

In this work, we investigated the retention characteristics of ICL in multi-task settings and extended it to ICCL through prompt scheduling and rearrangement. Experiments on benchmarks showed that ICCL benefits DP in a manner analogous to humans, consistently revealing a spacing sweet spot that enhances retention beyond SP and MP settings. Beyond retention performance, we introduced the HRS-MD metric to quantify alignment with human memory dynamics, and found that linear-attention models such as MAMBA and RWKV display particularly human-like retention patterns. Overall, our results establish ICCL as both cognitively plausible and practically effective, providing an inference-only paradigm that mitigates catastrophic forgetting and addresses the stability–plasticity dilemma in conventional continual learning methods.

## ETHICS STATEMENT

This work adheres to the ICLR Code of Ethics. Our study focuses on the development and evaluation of in-context continual learning methods using synthetic and publicly available benchmark datasets. No human subjects or sensitive personal data are involved. The proposed methods aim to improve retention dynamics in large language models without introducing harmful applications or privacy risks. All datasets used are either synthetic (Markov-chain sequences) or publicly accessible, and no proprietary or restricted data are employed. In addition, the large language models used in our experiments–LLaMA3-8B, DeepSeek-R1, MAMBA, and RWKV-7–are employed strictly in compliance with their open-source licenses. We are not aware of any conflicts of interest, sponsorship influence, or ethical risks associated with this research.

## REPRODUCIBILITY STATEMENT

We have taken multiple steps to ensure the reproducibility of our results. The main text provides a complete description of the proposed ICCL framework, the baseline methods, and the evaluation metric. Details of additional ablation studies are included in the Appendix. All datasets used in the experiments are synthetically generated Markov chain sequences or are publicly available. Hyperparameter choices and training configurations for both GBCL baselines and large language model baselines are fully documented. To further support replication, we will release our anonymized source code as supplemental materials submitted with the paper.

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

# A APPENDIX

## A.1 USE OF LLMS

We used large language models (LLMs) only as an auxiliary tool to improve the clarity and presentation of this paper. The assistance was limited to:

- **Language refinement:** grammar checking, wording suggestions, and improving sentence fluency while preserving the authors' original technical content.
- **Mathematical support:** verifying the correctness and readability of some derivations and notations, without introducing new technical results.

No LLM was used for generating research ideas, designing experiments, analyzing results, or writing original scientific content. All conceptual and technical contributions were made by the authors. The authors take full responsibility for the entire content of this paper, and LLMs are not eligible for authorship.

Table 4: Statistical reference means and standard derivations for human's ACT-R model parameters according to (Pavlik Jr & Anderson, 2005; Lewis & Vasishth, 2005; Bothell, 2020; Borst & Anderson, 2017; Said et al., 2016; van der Velde et al., 2022).

| Parameter | Mean value ($\mu$) | Standard derivation ($\sigma$) |
|---|---|---|
| Decay Rate ($d$) | $\mu_d = 0.50$ | $\sigma_d = 0.05$ |
| Activation Noise ($s$) | $\mu_s = 0.32$ | $\sigma_s = 0.08$ |
| Threshold ($\gamma$) | $\mu_\gamma = -0.50$ | $\sigma_\gamma = 0.71$ |

## A.2 ACT-R MODEL AND HRS-MD CALCULATION

To complement Section 2.3, we provide the full ACT-R retention model together with an intuitive explanation of its parameters and their roles in memory dynamics.

**ACT-R base-level activation and retrieval probability.** In ACT-R model (Pavlik Jr & Anderson, 2005; Lewis & Vasishth, 2005; Bothell, 2020), each practice creates a memory trace. If the learner is exposed to the practice at times $t_1, t_2, \ldots, t_k$, *memory strength* or *base-level activation* at a later time $t$ is:

$$w(t) \ = \ \ln \sum_{i=1}^{k} [\Delta t_i]^{-d}, \tag{7}$$

where $d$ controls how quickly each memory trace decays (larger $d$ implies faster forgetting) and $\Delta t_i$ is the time elapsed since the $i$-th exposure.

Memory activation is converted to a probability of successful recall using a logistic function:

$$P(t) \ = \ \frac{1}{1 + \exp\left(-\frac{w(t)-\gamma}{s}\right)}, \tag{8}$$

where $\gamma$ is the retrieval threshold (the minimum activation required for successful recall), $s$ is the activation noise controlling stochastic variability in retrieval. Together, equation 7 and equation 8 fully specify how ACT-R model predicts forgetting curves as a function of practice timing.

**Why Mahalanobis distance?** Human experiments report many $(d, s, \gamma)$ triplets, forming an ellipsoidal distribution with mean $\boldsymbol{\mu}$ and covariance $\boldsymbol{\Sigma}$. We evaluate the plausibility of the fitted ACT-R model parameters $\hat{\boldsymbol{\theta}} = [\hat{d}, \hat{s}, \hat{\gamma}]^\top$ with respect to human reference distribution characterized by $\boldsymbol{\mu} = [\mu_d, \mu_s, \mu_\gamma]^\top$ and a covariance matrix $\boldsymbol{\Sigma}$. Given $\boldsymbol{\mu}$ and $\boldsymbol{\Sigma}$, the squared Mahalanobis distance is

$$D^2 = (\hat{\boldsymbol{\theta}} - \boldsymbol{\mu})^\top \boldsymbol{\Sigma}^{-1} (\hat{\boldsymbol{\theta}} - \boldsymbol{\mu}),$$

which normalizes deviations by considering parameter scales and correlations. We report an interpretable similarity score $\text{Score} = \exp(-\frac{1}{2} D^2)$, interpretable as a Gaussian RBF under the Mahalanobis metric, to account for differences in parameter scales and their empirical correlations. Score measures how close a model's retention behavior is to that exhibited by humans. A small value indicates that the model's retention parameters fall within typical human ranges, whereas a large HRS-MD indicates atypical or non-human-like retention dynamics.

**Estimating the covariance matrix $\boldsymbol{\Sigma}$.** Table 4 reports the reference means and standard deviations of human ACT-R parameters from prior studies. Since only marginal statistics are available and raw human samples (hence empirical correlations) are not, we construct the covariance matrix from these reported scales. Let $\boldsymbol{\sigma} = [\sigma_d, \sigma_s, \sigma_\gamma]^\top$ denote the standard deviations and $\mathbf{D} = \text{diag}(\boldsymbol{\sigma})$.

By default, we adopt the independence approximation, assuming zero cross-covariances:

$$\boldsymbol{\Sigma} = \text{diag}(\sigma_d^2, \sigma_s^2, \sigma_\gamma^2).$$

This choice reflects the fact that no reliable correlation estimates are available and reduces the Mahalanobis distance to a multidimensional $z$-score distance.

## A.3 MODEL ARCHITECTURE AND PARAMETERS

**GBCL.** SGD, ER, and EWC employ the neural network structure. The neural network takes as input the current discrete state and a task identifier. By default, both are embedded through separate layers of size $\frac{d}{2}$ (with $d = 64$), concatenated, and passed to a multilayer perceptron with two hidden layers and ReLU activations: The output produces logits over the next-state space, which are transformed into probabilities via softmax. Weights of linear and embedding layers are initialized with Xavier uniform initialization, and biases are set to zero. Unless otherwise specified, we set $d = 64$, num_states$= 4/8$, and num_tasks$= 2$. Optimization in all cases uses stochastic gradient descent with cross-entropy loss. While the network architecture is identical across methods, their training hyperparameters differ. For SGD, the model is updated online using only the current sample with a learning rate of $0.001$. In ER, a replay buffer is employed with a capacity of $8000$ samples, a replay ratio of $0.5$, and a batch size of $32$. For EWC), the model is trained with the same online updates as SGD but augmented with a quadratic penalty weighted by the diagonal Fisher information matrix; the optimal regularization strength is set to $\lambda_{\text{EWC}} = 700$.

**LLMs.** LLaMA3-8B (Meta, 2024), DeepSeek-R1 (DeepSeek, 2024), MAMBA (TII, 2024), and RWKV-7 (RWKV, 2024) are used in their publicly released configurations without further fine-tuning. LLaMA3-8B is a transformer-based decoder model with approximately 8 billion parameters and a context length of 8K tokens. DeepSeek-R1 is a transformer model with roughly 7 billion parameters, optimized for reasoning tasks, supporting up to 8K tokens. MAMBA is a selective state-space model with about 8 billion parameters and an extended context length of 16K tokens. RWKV-7 adopts a hybrid RNN–transformer architecture, with around 7 billion parameters and a context length of 4K tokens. For all large models, we adopt the default inference hyperparameters recommended by their authors: greedy or temperature-controlled decoding with temperature in $[0.6, 0.8]$, maximum generation length matched to the context window, and no additional training beyond the released checkpoints.

## A.4 ICCL BENCHMARKS

**DMC benchmark:** To illustrate how a Discrete Markov Chain (DMC) prompt is constructed, we provide an example containing both the target task transition matrix $P_{target}$ and the interference task transition matrix $P_{interference}$. $P_{ij}$ represents the transition probability from state $i$ to state $j$. During prompt construction, samples are first generated based on $P$ to form the target task sequences, and later partially replaced by samples drawn from $P_{interference}$ to simulate interference phases.

Table 5: Example transition matrices for the DMC benchmark with total 4 states. $P_{target}$ denotes the target task transition matrix, and $P_{interference}$ denotes the interference task transition matrix.

| $P_{target}$ (Target Task) | | | | $P_{interference}$ (Other Task) | | | |
|---|---|---|---|---|---|---|---|
| 0.2694 | 0.2987 | 0.2681 | 0.1637 | $5.6 \times 10^{-8}$ | 0.50 | 0.50 | $5.6 \times 10^{-8}$ |
| 0.2161 | 0.0106 | 0.3997 | 0.3736 | $5.6 \times 10^{-8}$ | 0.50 | 0.50 | $5.6 \times 10^{-8}$ |
| 0.2489 | 0.2278 | 0.2645 | 0.2588 | $5.6 \times 10^{-8}$ | 0.50 | 0.50 | $5.6 \times 10^{-8}$ |
| 0.3021 | 0.2828 | 0.2460 | 0.1691 | $5.6 \times 10^{-8}$ | 0.50 | 0.50 | $5.6 \times 10^{-8}$ |

Examples of SP, MP and DP are shown below. Each prompt begins with the instruction *"Learn pattern, predict next state. [SWITCH_TO_INTERFERENCE] = interference pattern starts. [SWITCH_TO_NORMAL] = target pattern resumes. Predict next:"*, followed by a continuous sequence of sampled states. The control tokens `[SWITCH_TO_INTERFERENCE]` and `[SWITCH_TO_NORMAL]` mark the onset and offset of the interference period, respectively, allowing the model to experience alternating target and interference tasks within a single context.

> **SP prompt example (DMC benchmark)**
>
> ```
> Learn pattern, predict next state.  [SWITCH_TO_INTERFERENCE] =
> interference pattern starts.  [SWITCH_TO_NORMAL] = target pattern
> ```

resumes.  Predict next:

0131103010013130123312132023013310102001023312221001201221232223230 12
22313202232102022330320232012 03[SWITCH_TO_INTERFERENCE]21211222122112
21121111122212221221111111212221112111112122211221222221111122212111 1
11111121111122222222122222221212221121111222222112211221211121121211 121
22222211122111212211111221122121111111111212112221121112221121211212 211
12211121111212122112221122221211121112111121111222121222112221111111 11111
22211222121111111211122221212121111111211221122222211221221211111122 212
12121212211121112221122222111221111211122211211221212222111212112111 12112
21222121222122222111112211221222121111211122111221221211112222211122 2222211
21112221121222211122211221111111111121121112221122122222222221122222 12121
21222112221212111212211122211111112111222222222222211222111212112112 12122221
11111121112121211222112212222212221122221122221122122211112122111212 2222221
21211121221111212122122222221212121211111222121221121212121212112222 122
11221222212121112112222121112122112221212221211111112222112221121222 21
12222111221222212222111112111211112122112111121212122122221111112121 11121122112
22211121121122121112122112122212121212211211111121121121211121112112 2121
21222121211111112221211221212112112222121211111121111222212122121111 22122
22211121212211211211212121222122111122221222112122211111222212122211 1222
12121222112121121211111122222222111112122121212112211112222121111111 1121
22221111121112222112221221211212212112222111222211112222212221112112 11
11111112112212112111111221121112211111212211112111222212111222121111 212121
21112222212121211222212121122122221211222212222121122221122211121211 11
22112121221222222111112112112211211112211122221122112221112121211222 221
11222222112121222212222122212112112222122122122211212211221122121212 12111
12221112222222212222111111122121211111122221112121122121111112221212 22222
22121112211111221122122112121121212211222211121222212222221211121211 1121222
12222122212112121121112222222222221221222111112221222212212211222111 121
11221122221112212122212112212122121112112222122211122112111212112112 212212111
11221122112111222122122212221122121112111222122111111121212212222211 221
21222121211122212 2

MP prompt example (DMC benchmark)

Learn pattern, predict next state.  [SWITCH_TO_INTERFERENCE] =
interference pattern starts.  [SWITCH_TO_NORMAL] = target pattern
resumes.  Predict next:

0131103010013130123312132023013310102001023312221001201221232223230 12
22313202232102022330320232012033021222302200230131200012213313210001 0
12122312120000130332102202322331010123303010000121030000122323132023 3
32302022221312023132201320122121213002130122032322201023001301301123 2
01331201001010031301322123010231213020131330013312021200223023013230 1
33230310201301003012313130131202220121201210132300233130101213102313 3
13021312101020012201333231022032131012002320203020202212131022220223 2
212123100200332 00[SWITCH_TO_INTERFERENCE]21122121222211121211211121 12
21222121222122221111122112212221211121112211121221211122221112222222 11
21112221121222111222112211111111121121112221122122222222211222221212 1
21222112212121112122112221111112111222222222222211222111212112112122 221
11111121112121211222112212222212221122221122221122122211112122111212 2222221
21211121221111212122122222221212121211111222121221121212121212112222 122
11221222212121112112222121112122112221212221211111112222112221121222 21
12222111221222212222111112111211112122112111121212122122221111112121 11121122112
22211121121122121112122112122212121212211211111121121121211121112112 2121
21222121211111112221211221212112112222121211111121111222212122121111 22122
22211121212211211211212121222122111122221222112122211111222212122211 1222
12121222112121121211111122222222111112122121212112211112222121111111 1121
22221111121112222112221221211212212112222111222211112222212221112112 11
11111112112212112111111221121112211111212211112111222212111222121111 212121
21112222212121211222212121122122221211222212222121122221122211121211 11
22112121221222222111112112112211211112211122221122112221112121211222 221
11222222112121222212222122212112112222122122122211212211221122121212 12111
12221112222222212222111111122121211111122221112121122121111112221212 22222
22121112211111221122122112121121212211222211121222212222221211121211 1121222

```
1222212222121121211211112222222222212212221111122212222122122112222111121
1122112222112212122212112212221211211122221222111221121112121121121221211
1122112211211112222122122212221122121121112221221111111121212212222212211
2122212121211222122
```

**DP prompt example (DMC benchmark)**

```
Learn pattern, predict next state.  [SWITCH_TO_INTERFERENCE] =
interference pattern starts.  [SWITCH_TO_NORMAL] = target pattern
resumes.  Predict next:
0131103010013130123312132023013310102001023312221001201221232223232012
2231320223210202233032023201203312222100120122123222323012[SWITCH_TO_INTERFERENCE]1211222122112
2112111122212221221111111121222111211111212122112212222221111122212111
1111112111112222222212222221212221121112222221122112212111211121211121
2222221112211121221111221122121111111111212112221[SWITCH_TO_NORMAL]230
1023121302013133001331202120022302301323013323031020130100301231313013
3120222012120121013230023313012223130[SWITCH_TO_INTERFERENCE]1111121122222121
2121111112112211222222112212221211111122212121212122111211222112222211
1122111211222112112212122221111212112111211122122212122212222111122112211
2221211121112211122122121112222211222222211211[SWITCH_TO_NORMAL]232210
2133212131312310120010001310200222123122223132321013232130203131212122
2313123132103331001213100112131001[SWITCH_TO_INTERFERENCE]2222222222221112211
1212112122121111121112112212112222122212222112221222122121221122212221111
2122112122222212121121221111212212212222212121212111122212122112211121212
1212121122221221122122221212112112221121111[SWITCH_TO_NORMAL]303310031
3121232020001000322102330030332301222212032032321232210120301230102112
3300200131303202331213131[SWITCH_TO_INTERFERENCE]1112112211222211211222
1221211211121122121221122111112112222221122111111212122121212221221111
11122121122121211211222212121111121111222212122121112212222211212122111
2112112121212221221111122221222112212221[SWITCH_TO_NORMAL]12123233130233
0132313131333003012202030100003232221312102032132020300220123133130121
200130232312001301[SWITCH_TO_INTERFERENCE]122221122212221211212122121122
2211122221112222212221112112111111211222121121211111122112112211112122
1112111222212111222121112121212111222221212121122221212112212221222212112
2221222212112222112222112121211112211212121221222222111112112112211121111221
1122221122112221112121211222211122222211212122221222212221211211222212
2122122211212211221122121212111122211122222221222211111112221212111112
2221112121112211211111222121222222221211221111221122122112121112122112222
1112122212222221211212111121222122221222121112112112111122222222222212212
2211111222122221221221122211112111221112221122121222121122212212112112
222122211221121111212112112212121111122112211211112222122122221222112212
1211112221221111111212122122222122212122212121211222122
```

**REGBENCH benchmark:** We also construct a symbolic sequence benchmark based on RegBench, where both the target and interference tasks are generated by probabilistic finite automata (PFAs) implemented as deterministic finite automata (DFAs). Concretely, we first sample a target DFA and an interference DFA with different numbers of states, alphabet sizes, and random seeds. Each DFA induces a stationary stochastic process over a finite alphabet, and we generate short examples by running the corresponding DFA for a random length between $L_{\min}$ and $L_{\max}$ symbols. We concatenate such examples into a single long sequence and insert control tokens [SWITCH_TO_INTERFERENCE] and [SWITCH_TO_NORMAL] whenever the generator switches between the target DFA and the interference DFA. This yields a context in which the model must continuously track the active PFA and predict the next symbol in the sequence.

SP, MP and DP examples are explained as follows. In our setup, the target task is produced by a DFA with *4 states*, *alphabet size 10*, *maximum 2 outgoing edges per state*, and a fixed random seed (seed=42) to ensure reproducibility. The interference task is generated by a distinct DFA with *6 states*, *alphabet size 20*, and *maximum 3 outgoing edges per state*, ensuring a substantially different symbolic distribution and transition structure. Each DFA defines a stationary symbolic process, and we generate short examples by running the corresponding automaton for a random length sampled from $L \sim \mathrm{Uniform}(10, 15)$. We concatenate $N_t = 320$ such examples to form a prompt. We additionally generate a separate test split consisting of $N_{\text{test}} = 10$ pure target-DFA examples, whose

lengths are sampled from Uniform(8, 10). This test set assesses how much target-pattern ability the LLM retains after experiencing last historical demonstrations in a prompt.

---

**SP prompt example (Regbench benchmark)**

```
Learn pattern from probabilistic finite automate(PFA) sequence.
[SWITCH_TO_INTERFERENCE] = interference pattern starts.
[SWITCH_TO_NORMAL] = target pattern resumes.  Predict next:  m n m n m n
mnmssw,mssmswmnmsswmn,msswmssmsw,mnmssmsmswmssw,mssmsmswmss,mssmsw
mssmswmss,mnmsswmnmnm,mnmssmsmsms,mnmssmsmswmnm,mssmsmsmsw,mnmsswm
nmnmsswm,mssmswmssmswmss,msswmssmsmswms,mnmnmsswmssm,msswmsswmn,mnm
nmssmsm,msswmssmsmsmsms,msswmnmnmnmn,msswmnmsswmnmss,msswmnmnmsswm,
[SWITCH_TO_INTERFERENCE],  nvemxxdrux,dpaxuqrtem,dxdrdnmqrure,dxuqnmqd
pe,rtemxpevevaxdn,ngdpegdxdn,ruxpapaxuqr,nvapaxtevaxddxd,ngngngrteve
mra,dpaxdnvegdx,rtaxtevaxura,nvegrtapeve,ngrdruqrtemx,ngrtapaxuqdx,nv
emqrtaxure,ruqdxuqngnmrax,nmxpaxddxdnv,rtaxuxpevapaxd,dxdruqdxurapemr
,ruqnmregrtev,dxtevaxurapa,rdnvemremxxddp,ngdpaxdngng,dpaxtemqdpevem
,dpegnmregrdr,dxdngnvapevap,ruxxtaxureveva,ngruqdpemregrdd,dxtaxddpev,
nmqrddpegnveg,nmxpegnmxp,dxurapegdpemxp,dxurevevaxdngrd,rddpevemregnv
,dxdngdxdngru,rdnmxxdrdrte,ngdpapapapemq,rtapevegdpapeg,rurapemxpegruxx,
dxuqdpegnva,dxtapapegngdx,nvegruqdxdnvape,dxtegrtegnmqdpe,ruxxuxxtemq,
ngrtevegnmra,dpaxuqruxxdnvax,ruxpemraxddp,rddxuxxuqnmrev,nvegrtapegdx
ddp,dpapemrapegdpap,nmrevemraxuxpev,dxdrdngdxd,rtegdpemxxur,nmqngruqn
mxpem,rdrdnvaxddxdd,dxddpegdpapaxur,dxdruqrdrura,nvemrevapaxu,dpemrem
xxuqngrd,nvemqnmrevap,nmqdpapape,nmraxddxtap,rtaxurapevemxp,rtaxuxxdr
teveve,ngdxtaxtegruqrt,rtapaxtaxta,nvapapapapapeg,dpegdpemqdpa,dpegnm
rape,dxdrdnvapaxtev,dpaxuxpapa,rtemrevemqnmxx,dxdnmqnmxpapa,ruqngrdng
rdnm,nvapaxurapaxd,dxtapaxuqrddx,ngngnmqdxddpa,dxuxxddpaxdr,nmqrdrdru
remxx,ngrtemremr,nvevevevapapa,nvapevegrt,dxuqrtegnvevegd,rdnvevaxtap,
ngnmregnmq,dxtegdxuxpa,ngruxpevegdxura,dxuqdpemrapegru,dpevemqngruqnm,
nmqnvemxpe,nvegdpaxdn,ruraxddxdng,dxddxtemxpapemx,dxurapaxddxddxu,dxu
qdxuqngn,ngdxdrurevegdx,dpaxdrdrdn,dpemrapape,dxuremxxurem,dpevapaxuq
rdd,rdruraxureg,nvaxtevapapa,nvevemqruqnveg,rddpapaxddpap,dxuqdxtapax,
dxtapaxurapax,ngruqrdrta,ngngrddpeg,dpegnvaxurax,ruqdxdrddxd,rtemxxta
xtaxdn,rdruraxdruregdx,nmqngruxpa,ngdxtapaxtaxtap,ngnvegdpap,ngrtapev
axteve,dxtevapegnmxpe,dpaxuxxddpeg,ngdxtemqdpegnmq,nvapegdpaxur,rdrux
papemrem,dpapevaxte,dpemraxuqnvap,ngdxdruxxdn,nvapevevevapemx,ngnmreg
rteg,dpaxuqrdruxxdnm,rtevevevevegdxd,rdrdrddxtegrdr,nvemqnvapapegdp,r
tevemreva,nvaxuqruqdpe,ngruxxuxpapemr,nvevegrdruxpe,dxddxtevapaxur,ng
nmqrtegdpapem,ngdxddxdnvevax,rddpegnvaxuremq,nmxxtevaxdnveve,dxtegdxd
ru,nmraxdnvegnmr,ngdxddxdruxpape,nvaxuqnmqr,nmqnmxxuxpevax,nmremqrtev
axu,dpaxtapapemra,rdngdpevap,rtaxdrtemqdxta,ruregrurapegdxu,dpevaxure
v,rtaxuqrtapegdx,rdngruxxdrdngr,dxtaxdngdpemrem,dpaxddxddpaxd,dpapape
gnvax,rdngrddxuraxdd,rddpaxurap,dxuxpapaxdr,dpegngdxuqrdn,nmrevaxdngr
tem,dxdnvemxxdrur,dpevegnvevapapa,rtegrtemqrtapa,rtevaxuxpaxu,rtemqdp
apaxt,nmqruqrurapemq,nvaxuremxpeg,nmxpaxtemxpa,ngruqdxtegr,nmqngnveva
x,rurapemrape,rdrtapaxtapaxu,dxtapemxxdnm,rtapaxuqnmraxux,dpaxddxuxxt
e,nvegngdxta,rdngdxddxuxp,nvaxdrurapaxdrd,nvapevemraxtev,dpapemqnvapa
,dxtevapaxt,rtaxtegrdr,nvevemxxtaxtem,dxuqngdxdngng,nvapemremra,ruxxt
apevev,ngngngnmra,dpaxuxpevap,dxdngnvegdpe,rtegruraxtegrt,ngnmqrtemqr
ur,dxuremraxdrdngd,dpegdxuqrdnm,dxtapevapemqn,dxuqdpaxdn,ngnvegnmxpem
qnm,ruxpegnvaxdrd,rdrtemqruxpaxte,rddxuremxpax,rtevaxuraxdr,ngngrtegn
gdxu,ngdpevapem,dpegruqruqdpax,rdnmqdxuregrte,rtemqdpevevaxu,rddpegdx
dnveve,dxuxxtegrddpemq,dxddpevapape,nvemxpapeg,nmqdpapapegn,nmraxurem
regrt,nmrapapapapem,ngdxdnvaxte,ngngngngnmraxdn,dxtapaxdnmx,dpemraxur
ax,rddpapapevax,nvapevaxddpax,nvegrdrtegru,dxtaxtapaxuregn,dpegdxtegr,
nmraxddpapeva,ngruqngrur,dpemraxuraxdrdn,rdnmxpaxdd,dxuxxtapaxdrd,ngd
pevapegnvax,nvemrapapegrt,nvaxuxpevaxtax,dxtaxurapapev,dpegdxdnmq,dxd
rtegdpaxu,nmraxtapeg,ngrtemqdxtemxp,nvaxuxxuxpe,dxdnmrevaxd,ruxpemrax
u,ngdxdnmqrtev,dpemqngruqd,rurapaxtaxte,dxtemraxdd,dpemqnmremqnmra,dp
egnveveg,dxtevaxtemqrd,nmqnvaxuxpaxu,rdrddpevaxddxd,nvegrurevax,dxtax
tegruqru,ngrtemregrtapa,rdngdpemxxtapa,rddxdruraxdrure,dxddxddpax,ngn
gdxdnmreg,ngruxpapaxuq,rtegngngdpem,dxdrdrdruqdxd,dxdrtapegdxuxpa,ngr
taxdnva,nvaxdrtaxtegrta,rdrtaxtaxu,rdngnmxpegdpape,dpaxuraxtaxtemr,rd
```

```
ruxxurapapem, rdngrdngrdnva, rddpaxuxxu, ruregdxddxtap, nmxpevapegrdrdn, n
vemxxddpemx, ruxxdruxxd, dpapegngrdrte, dxdnmqrtegrtapa, ruqdpaxtemrape, d
xddpaxuqrta, dpaxuqrddpe, nmrapemqruqd, nvegrdnmxxdrtem, dpaxdruqru, nmqdx
drdngdpemr, dxtemraxddpa, dxdnvaxuxxura, ruxpemxpaxuqdpe, nmremqnmremx, dx
ddpapaxtap, dpegdpemqrteve, dxdrurapegrd, ngdxtegruqrdrd, ruxpapegdxta, rd
dxtevaxdn, ruraxuqnvaxuqng, dpaxtevapaxta, dxdnmxxuraxdd, ngrtaxtaxdngngd
, nmqruqnmxpape, nmxxtevaxura, nvevapemqruqd, ruxpevevegdxuq, ngdpevevemqd,
rddpapemxxu, nvapemremr, dpevevemqru
```

```
Learn pattern from probabilistic finite automate(PFA) sequence.
[SWITCH_TO_INTERFERENCE] = interference pattern starts.
[SWITCH_TO_NORMAL] = target pattern resumes.  Predict next:  m n m n m
mnmssm, msswmnmnmssmsm, mssmswmssm, mnmsswmssmswmn, msswmssmswm, msswmnms
swmnmss, mnmssmsmsms, mnmsswmsswm, mnmsswmssmsms, msswmsswmn, mnmssmsmsms
wmnm, msswmnmsswmnmss, mssmswmsswmnms, mnmnmssmswms, mssmswmnmn, mnmnmssw
ms, mssmswmsswmsswm, mssmsmsmsm, mssmsmswmnmnmss, mssmsmsmswmnm, mnmnms
swms, msswmnmnmn, mnmnmsswmssm, mssmsmsmsm, mssmswmsswmssw, mnmssmswms, ms
smswmssms, mnmnmnmssmsmsms, mnmssmswmnmnms, msswmnmnmss, mssmsmsmsmsw, mn
mnmsswmss, mnmsswmsswmn, mnmsswmnmnms, mnmnmsswmnms, msswmnmssmswmn, mnms
swmssmsm, mssmsmsmswmssw, mnmnmsswmssmsms, mssmsmsmswmn, mnmnmnmssmsm, ms
swmnmnmsswms, mnmnmnmssms, mnmssmsmsmswms, msswmsswmnms, mnmnmssmsmsms, m
ssmswmnmnmnmn, mnmnmssmsmsmswm, mssmsmsmsm, mnmssmswmssms, mnmsswmssw, ms
smswmnmnmssw, mssmswmsswmnmss, msswmnmsswmnm, mnmnmssmswms, msswmnmsswmn
, mnmnmsswmnm, mssmsmswmssmsw, mssmsmsmswmssw, mssmswmnmss, mssmsmsmswmnm
, mnmnmssmswmnmnm, mssmsmswmnmssms, msswmssmswm, mnmsswmnmssm, mnmssmsmsm
smswm, mssmswmsswms, msswmssmsmswmn, mnmnmssmswmnmnm, mnmssmswmsswmnm, mn
mssmswmnmsswm, mssmswmssw, mssmswmnmssw, mnmssmsmswmssw, msswmsswmnmnm, m
nmnmssmswmsswm, mssmswmssmsm, mnmnmsswmnmn, mnmssmswmsswms, mnmnmsswmnm
n, mnmssmsmsw, mnmssmsmsms, mssmsmsmsmswmn, mssmsmsmswmssms, mnmssmsmsmsm
sms, mssmsmswmnm, mnmnmnmssmswmn, mnmsswmnmssw, msswmsswmn, mssmswmsswmnm
n, mnmssmsmsw, mssmswmssmswmn, mssmswmnmssms, msswmnmnmnmss, mnmnmnmssmsm
s, mnmnmnmnmssms, mnmnmnmnmnmss, mnmnmnmnssmsw, mnmssmswmsswms, mnmnmssmsw,
[SWITCH_TO_INTERFERENCE],  dxtegdpemxxte, dxdruqnvax, rdrdnmraxdrdrur, ng
dxuqdxdrt, dxdnvaxdnm, ruqrurevape, dpaxtapegruraxd, rurevapaxuxpapa, rtax
taxdrtemqn, dpemqrtapa, dxtapaxdnm, nvaxdnvegdp, rdnvemxpapegngn, rdruraxu
xpegrur, rdrdrdrddpe, dxuqngruxxurem, ruqdpapape, rtapemregn, rdruxpaxurap,
rtaxuxxuregnv, ngnvaxdruxx, dxddxtemxxuq, dxtevaxtaxddpa, ngrtegruxxura, r
urevemqruq, rddxtevevevem, dxtaxtapem, dpevapapax, rtapemqdpaxd, nvevegngr
dn, nmxpaxddxdruxx, ngnvemqngruxxuq, dpemraxtax, dxuqdxtemqdxddp, dpemxxux
xt, dxtemxpemregdp, rddpevapaxtapa, ruqdpaxtegnv, dxuqdpaxuxpaxte, dxdrure
grure, ngnvapegrteva, ruqrtemqdp, rtapemremqdxt, dpemxpegdpe, dxuxpevegdxt
apa, dpegrtapegr, rtemregngrtegdp, nmxxtegdpeveveg, ngngngruqnvapa, dxtaxd
dxtegrura, nmxxtapaxu, dxuqrdnvevap, dxtapevapegrte, dxuqnvapaxtap, rdnvem
xxdruqru, dxdnmxxuregdpap, dpemxxdnmqnmre, ngrtaxddxuregnm, dpaxdnmregdxt
ev, ruqnvaxtax, dpevegdxuregr, dxurapegnvapegr, dxuremqnmx, dpemxpevapaxur,
dpapaxtegdxdr, rtemqdpegngrd, ngdpapaxdr, nmregnmxpevemq, nvapapevevapaxu,
ruxxddxure, nmqruqnmqrurax, ngdxtegdpapeva, rddxdnmrapapegn, rtemxxuxxtem
x, rtegrtapemqd, ngdpapemraxdnv, ngrtemremr, rdrtaxtemxp, ruxpevaxuqngd, dp
apevegdxteva, ruxxdrtaxtaxu, rtaxtaxdnmqnvax, nmrapevaxtevax, nmxxddxtaxd
r, nmraxuxxtemq, dpegrdnvaxurax, dxddxuxpapap, dpapemqngngr, dpapevemreg, d
pemremxxdr, nvaxuxpemxp, ngnmqnmqdxtemr, rddxtapaxteg, nmraxdrddpaxdrt, rt
emxxuraxdr, dxurevaxdr, ngdxuqnvevap, dxddpaxuqnmqngn, dxdrtevaxdrdnm, rte
grtemqruq, ruqrdrtemq, nmqdxtegng, dxtegngrdruqrt, rdrddpemxxuqr, dxdngnva
pax, nvaxddpaxte, dpevevegrd, rtemrapaxdn, rdnmremraxta, nmxpapaxuqnvax, dx
dnmxxuxxtax, rdruxpaxdngnmra, rtaxuqrdngdp, rddxuremxpaxd, rdrdruqdpe, dpe
mraxtapapemx, nvapaxddxdngn, ngnmraxtegnmqdp, ngrdruxpapem, nmxxdrurevap
, dpevaxurevemr, dpegrdrtap, rtapapegrdrtem, ngdpevevegrtev, nmxpevapegdxd
r, ngruxxuraxdrdn, rdrtemxxtapapax, ruxxuxxdnmxp, dxtapaxuxp, dpevaxtegnve
, dpevevapapaxte, dpaxuraxtevap, dpemxxddxdr, dxdrdrdrdnvemxx, rddxuregdpa,
rtapevevemq, ngruqnvapemq, dxuraxddpevem, dxurapaxurap, rddxuqrtemrevem, r
tapemqrux, dpaxuxxtegnmq, dpapemraxu, rtapemrapemxpax, ngdpapemxx, ruraxdd
```

```
pemxpa,dxuxpemrapemqd,dxuxxuxxtapax,dxddxtapemqdxd,rddxdruraxure,ruxx
uqnmxx,rdngdpevaxdr,dpaxddxtax,dxtegnmremxpap,dxddxtevapa,rdnmxpaxddp,
nvapapemra,dxuraxtegdpe,ruraxdrtaxu,nvaxuqrdruqr,ruqrtevegr,rtapemxpa
pemxpe,rtapemregr,rddpevemxpegn,dpemqdxuxpemr,ngngrtaxuraxdn,dxtegrur
emq,rddxddpapemxx,dpaxtapaxtevax,ngdxuxpapemrax,ngruxpapevapeva,rdnve
grtem,dxdruqnmxpap,dxtapaxuqdpe,nmxxdrdruxpa,rdngngngrdrdn,rdngdxurax
drtax,dpaxuraxdr,dxddpaxddxtegdx,ngnmqrddxu,ngdxdngngruxxta,rtemremre
mregnv,ngnvaxurevegng,ngdxtaxuxpemq,ngrtemraxu,nvegruraxuqdp,dpapaxdn
grtapax,dxtapegruxpa,nvaxtapaxt,rtegruqrtapev,rdnmxxtegrtevax,nvevaxu
qngrdru,rdnvaxdrdnmr,ruqdpegnvap,dpaxtapegrdr,dxtapaxtaxtaxux,rtemxpa
xta,dpevegngdxuxpax,rddpapemxxur,ruxxddxuxxure,nvapapapemrevap,dpegnm
qnvapa,rdnvaxuremqn,ruxpapapegdpev,ruxpevevapap,dxuqdpegrdngng,nvapeg
ruremq,ngrddpemqnm,nvemqrddxdruqdp,rtemqnmqrddxd,rdnmxpevevegr,dxtemr
emqnmreve,dpegrddpapegn,dpaxdnmqdxur,dxtevapaxtaxu,nvapaxtegruqrd,dpe
gnmxxtaxu,ngrtegngnmr,dxdnvapapa,rtaxtegnmxp
```

---

**DP prompt example (Regbench benchmark)**

```
Learn pattern from probabilistic finite automate (PFA) sequence.
[SWITCH_TO_INTERFERENCE] = interference pattern starts.
[SWITCH_TO_NORMAL] = target pattern resumes.  Predict next:
mnmnmnmnmssw,mssmswmnmsswmn,msswmssmsw,mnmssmsmswmssw,mssmsmswmss,mss
mswmssmswmss,mnmsswmnmnm,mnmssmsmsms,mnmssmsmswmnm,mssmsmsmsw,mnmsswm
nmnmsswm,mssmswmssmswmss,msswmssmsmswms,mnmnmsswmssm,msswmsswmn,mnmn
mssmsm,msswmssmsmsmsms,msswmnmnmnmn,msswmnmsswmnmss,msswmnmnmsswm,mnm
nmssmsm, [SWITCH_TO_INTERFERENCE], nvaxuqdxdd,nvegdxdrdnve,nmraxdruxp,
dxuxpegruqnmxx,rurevapapa,dxddxdrtemx,rtapaxtemrapemx,ruqnmremrapeva,
ngnvemqdpem,dxtaxdruqnva,rtevemqdpap,ruxpevapapeg,ruxxtemqnmra,rtegrd
rtapap,dpapapaxuqdpax,rddpaxdnmxxt,dxuqngnvaxurap,nmxpemrapemxpev,dxt
apapegrdr, [SWITCH_TO_NORMAL], mnmnmnmsswmn,mssmswmnmssmsm,mnmnmnmssw
m,mnmsswmnmnmssm,mssmsmsmswms,mnmnmsswmnmnm,msswmsswmnmnmn,mnmnmsswmn
mnmss,msswmnmnmn,mnmsswmssmswm,mnmssmsmsm,msswmsswmnmssm,msswmssmsmsw
mss,mssmswmssmswm,mnmnmsswmssm,mssmswmssmsw,mnmnmssmswm,msswmnmssmswm
s,msswmnmnmssmsms,msswmsswmss,msswmnmnmsswm, [SWITCH_TO_INTERFERENCE],
rtevevaxtemqngd,nmqnvemxpapapeg,dpemrevapem,ruxxdrdnmxxu,nvaxuqdxdrd
rtax,dxddpevaxdnv,dpemremraxdngr,rtevemregnvapap,nvapegnvegnvemr,rd
nvapevaxuxpem,nmxpemraxd,dxdnvegdpape,rdrtegnmqdpegd,dpegrtaxdnvev,n
mxxddpapapaxur,nmxpaxddpaxt,rtegngrtemqn,ngdpapegnvaxuxp,rtegrtevemq
d, [SWITCH_TO_NORMAL], mnmsswmnms,mnmnmsswmnmnm,msswmnmnmnmssw,msswmnmn
mssmswm,mnmsswmnmnmnmnm,msswmnmsswm,mnmnmnmsswmssw,mnmssmswmssm,mssm
smsmsw,msswmssmsmswmn,mnmsswmnms,msswmssmswmssw,msswmsswmsswm,mssmswm
nmnmss,mnmnmnmsswmnm,mnmnmnmnmsswm,mnmnmnmnmnmss,mnmnmnmsswms,mnmssw
mssmsmsm,mnmnmsswms,mnmnmsswmsswm, [SWITCH_TO_INTERFERENCE], rtapemxx
dr,nmrapapemqnmrax,dpaxuqrtaxd,ruqdpevapa,nvapapapegn,ruxxdnvapaxura
x,nvaxuxpevapevev,ngdxddxtapaxte,rdrtapegng,rtevegnmxx,dpemqdpapap,n
mxxtapegnvegdp,nmrevemrevapaxu,nmraxtaxtem,rurapevapapeva,nvaxddpegr
,ngdpaxureg,nmrapegrureg,ngdxuxxurapev, [SWITCH_TO_NORMAL], mssmsmswm
ss,mnmnmnmnmnms,mnmnmsswmssmsw,mssmswmsswmss,mnmnmsswmss,mnmnmnmnmss
wm,mnmnmssmsw,mnmsswmnms,mnmssmsmswmn,mssmswmsswm,msswmssmsmswmn,mss
msmswmnmnmnm,mnmsswmnms,mnmnmnmnmnmnmss,mnmsswmnms,mnmnmsswmnmssm,mn
mnmnmsswmnms,mnmsswmnmssm,mnmnmnmnmnmssms,mnmnmnmssmsw,mssmsmsmsmsms,
[SWITCH_TO_INTERFERENCE], nvapemqrte,ngngnmraxtaxd,rurapevemxx,rtapem
rapaxddpe,ruqdxdngrdn,ngrtaxddpegngrd,dxdrdrdrdruremx,dpegdpaxtevegd,
rtegrtaxdngnveg,dxdrddpemr,rtaxuqdpaxux,ruxpemrevapegn,rtemxpegdxddp,
nmxpaxdrtevaxu,ruqdxdrurapapeg,rurapaxdrtemra,dpapevaxuraxddx,rddxtem
rapaxdru,nvapapapev
```

### A.5   RETENTION PERFORMANCE UNDER SP, DP AND MP FOR DMC REGBENCH

Figure 6 illustrates the retention performance of ICCL models and GBCL baselines varies on the simple task with task identifiers and $\varphi_I = 200$, under SP, MP, and DP scheduling. For ICCL methods, we observe that in both SP and MP conditions, the retention performance drops sharply immediately after finishing all target task demonstrations, and subsequently remains at a reduced level as the

sequence length increases. In contrast, ICCL under DP consistently achieves better performance when the sequence length reaches its maximum, demonstrating its advantage in retaining cross-task knowledge. However, for GBCL baselines, DP does not yield a comparable improvement and performance suffers sharp drops across sequence lengths. These results confirm that both ICCL and GBCL models suffer from catastrophic forgetting, yet DP is effective in mitigating this issue for ICCL by alleviating performance degradation over long sequences.

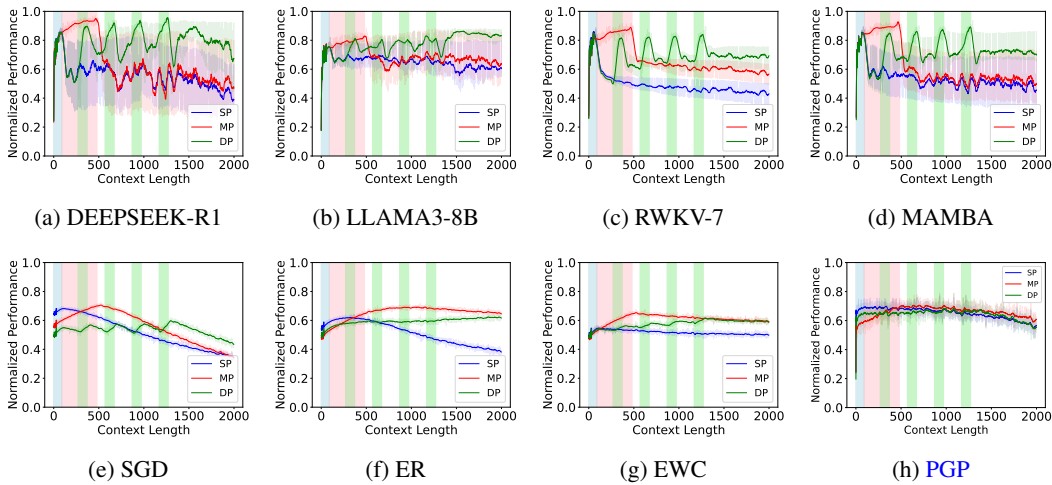

Figure 6: Retention performance of ICCL and GBCL models on simple task under SP, MP, and DP settings (with task identifiers, $\varphi_I = 200$). Target task blocks are highlighted in light blue (SP), pink (MP), and green (DP).

## A.6 RETENTION PERFORMANCE UNDER SP, DP AND MP FOR REGBENCH BENCHMARK

Figure 7 presents the retention performance of ICCL models on the RegBench task with task identifiers ($\varphi_I = 20$, $\varphi = 40$, $K = 5$, and $\varphi_D = 120$) under SP, MP, and DP scheduling. Consistent with the observations on the DMC benchmark, all ICCL methods under SP and MP experience an immediate and pronounced performance drop once the demonstrations for the target task conclude, after which their accuracy stabilizes at a lower level as the sequence length continues to grow. In contrast, under DP scheduling, ICCL models maintain consistently strong performance without exhibiting the characteristic sharp decline even when the sequence reaches its maximum length, highlighting DP's advantage in preserving cross-task knowledge. Evaluating ICCL methods on both DMC and RegBench benchmarks further demonstrates that ICCL is effective not only in purely synthetic settings but also in language-style sequence environments, indicating strong generalizability across heterogeneous task formats.

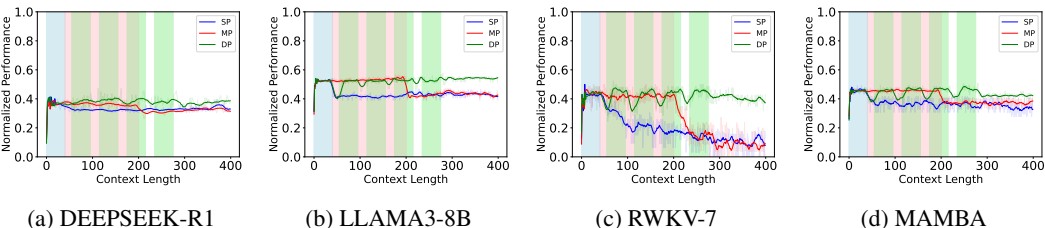

Figure 7: Retention performance of ICCL models on regbench benchmark under SP, MP, and DP settings (with task identifiers, $\varphi_I = 20$). Target task blocks are highlighted in light blue (SP), pink (MP), and green (DP).

Table 6: The comparisons between the true normalized performance of ICCL and GBCL methods and their fitted ACT-R curves.

| Model | DEEPSEEK-R1 | LLAMA3-8B | RWKV-7 | MAMBA | SGD | ER | EWC |
|---|---|---|---|---|---|---|---|
| Pearson correlation | 0.5152 | 0.7068 | 0.6171 | 0.6215 | 0.0339 | 0.5660 | 0.7253 |

## A.7 OPTIMAL PARAMETER $\lambda$ IN EWC

Figure 8 and Figure 9 present the performance of EWC under simple and complex tasks with different regularization strengths $\lambda_{\text{EWC}}$ across SP, MP, and DP scheduling strategies. The results highlight the trade-off between plasticity and stability inherent in EWC. With small $\lambda_{\text{EWC}}$ values (e.g., $\lambda_{\text{EWC}} = 100$), the model maintains higher plasticity and adapts quickly to new demonstrations; however, this comes at the cost of poor stability, as retention performance decays sharply with increasing context length. Conversely, large $\lambda_{\text{EWC}}$ values (e.g., $\lambda_{\text{EWC}} = 1000$) enforce stronger stability but severely limit plasticity, leading to under-adaptation and overall lower performance across all scheduling strategies.

Intermediate values achieve a better balance. In particular, $\lambda_{\text{EWC}} = 700$ consistently provides a favorable trade-off, delivering higher long-term retention compared to smaller $\lambda_{\text{EWC}}$ while avoiding the rigidity observed at $\lambda_{\text{EWC}} = 1000$. This effect is especially evident under MP and DP, where $\lambda_{\text{EWC}} = 700$ maintains relatively stable retention curves while smaller $\lambda_{\text{EWC}}$ values degrade more rapidly. Based on these observations, we select $\lambda_{\text{EWC}} = 700$ as the default regularization strength for subsequent experiments, as it offers the most robust balance between plasticity and stability across both simple and complex tasks

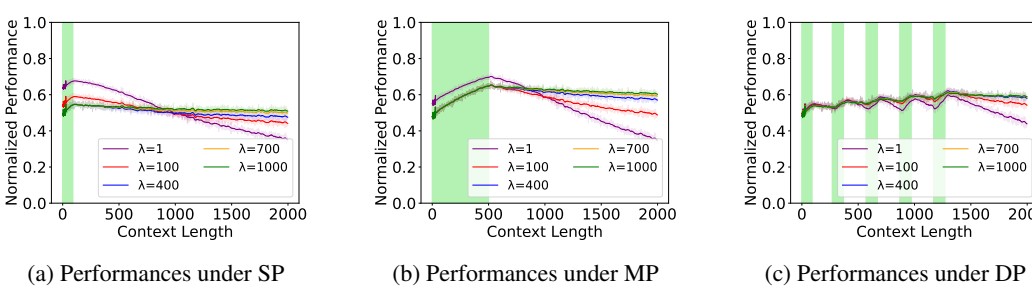

(a) Performances under SP  (b) Performances under MP  (c) Performances under DP

Figure 8: EWC's retention performances under Markov chains with simple task (markov chain with $Ns = 4$).

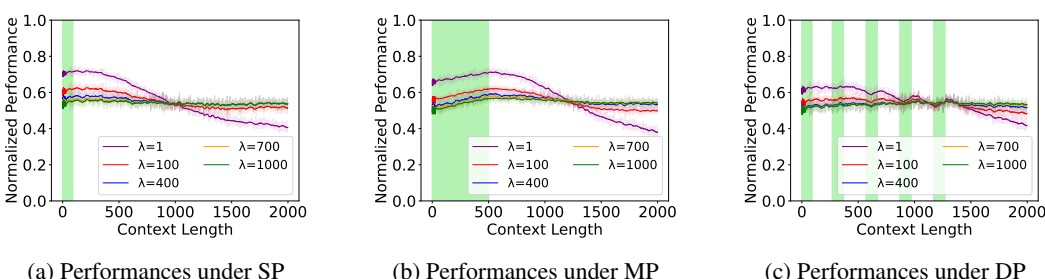

(a) Performances under SP  (b) Performances under MP  (c) Performances under DP

Figure 9: EWC's retention performances under Markov chains with complex task (markov chain with $Ns = 8$).

## A.8 THE FITTED ACT-R MODEL RESULTS FROM ICCL AND GBCL

To further analyze the retention dynamics of ICCL and GBCL methods, we leveraged the experimental data of normalized retention performance across prompt sequence positions under the DP setting. We

applied a modified ACT-R model to fit the performance trajectories of each method. The comparisons between the observed normalized retention performance curves and the corresponding fitted ACT-R curves on both simple and complex tasks are illustrated in Figures 10–16.

In addition, we computed the average Pearson correlation coefficients between the true normalized retention performance curves and their fitted ACT-R counterparts, as reported in Table 6. The results show that, with the exception of SGD, all ICCL and GBCL methods achieve relatively high correlation coefficients (e.g., 0.71 for LLaMA3-8B and 0.73 for EWC). This indicates that the fitted ACT-R model provides a reliable description of the dynamics of normalized retention performance, capturing how ICCL and GBCL retention evolves with increasing sequence positions.

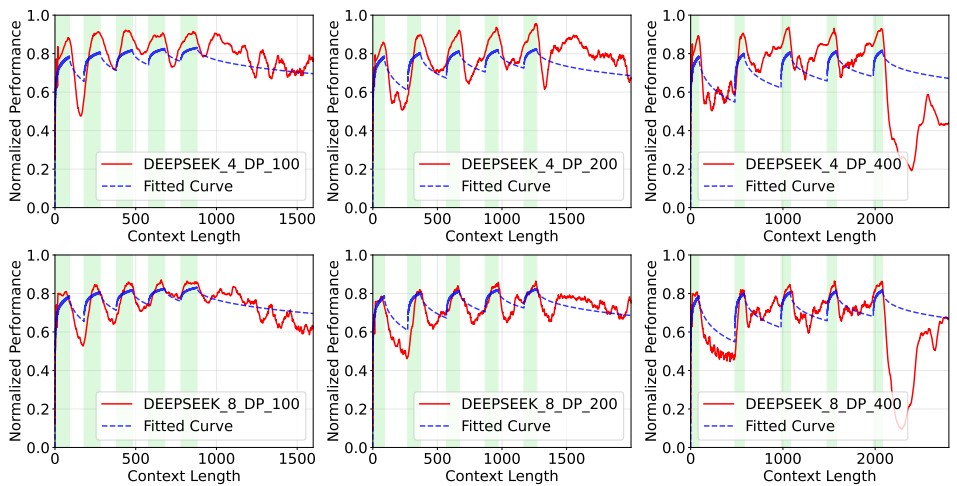

Figure 10: The performance comparisons between DEEPSEEK-R1 and its fitted ACT-R model.

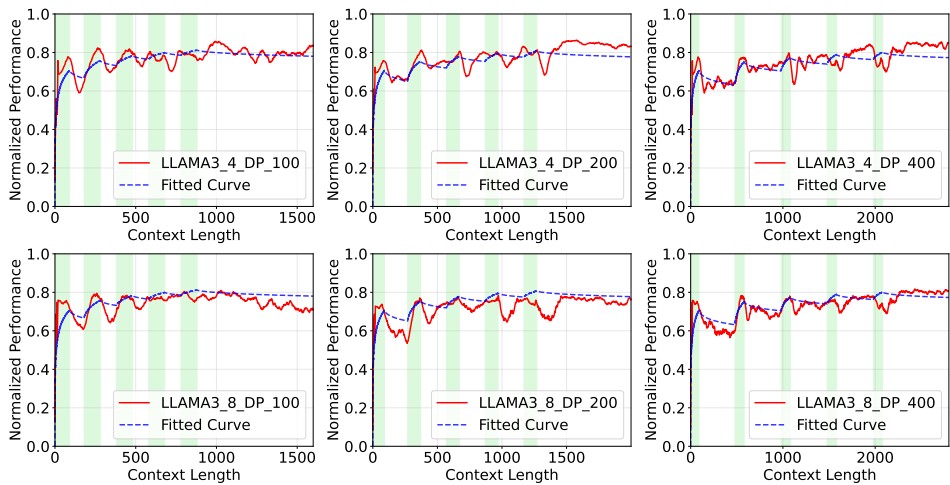

Figure 11: The performance comparisons between LLAMA3-8B and its fitted ACT-R model.

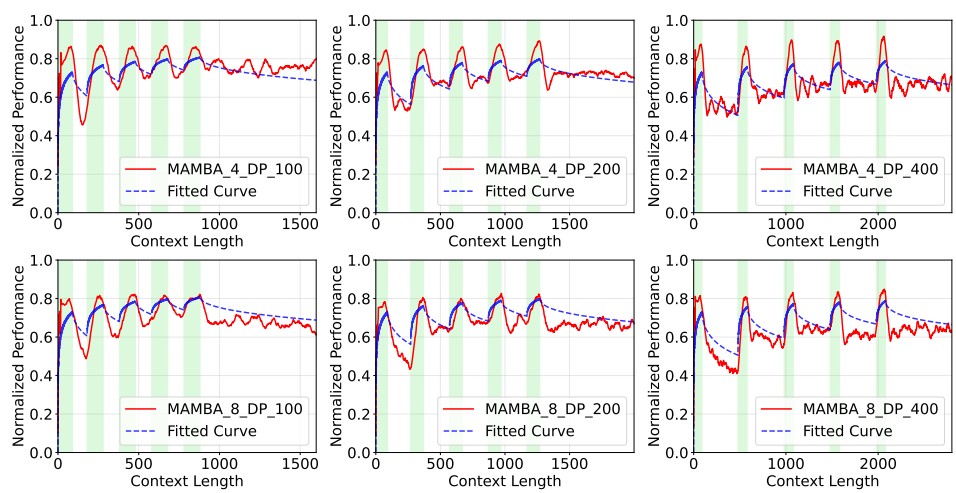

Figure 12: The performance comparisons between MAMBA and its fitted ACT-R model.

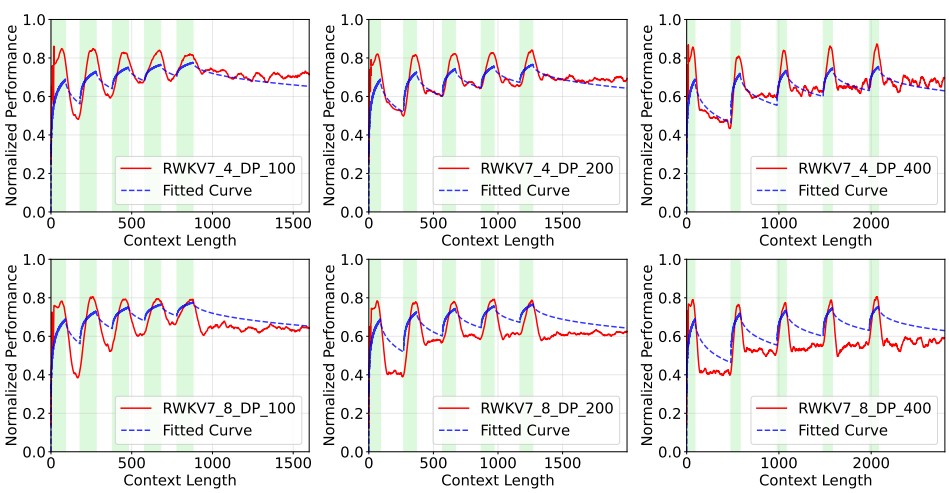

Figure 13: The performance comparisons between RWKV-7 and its fitted ACT-R model.

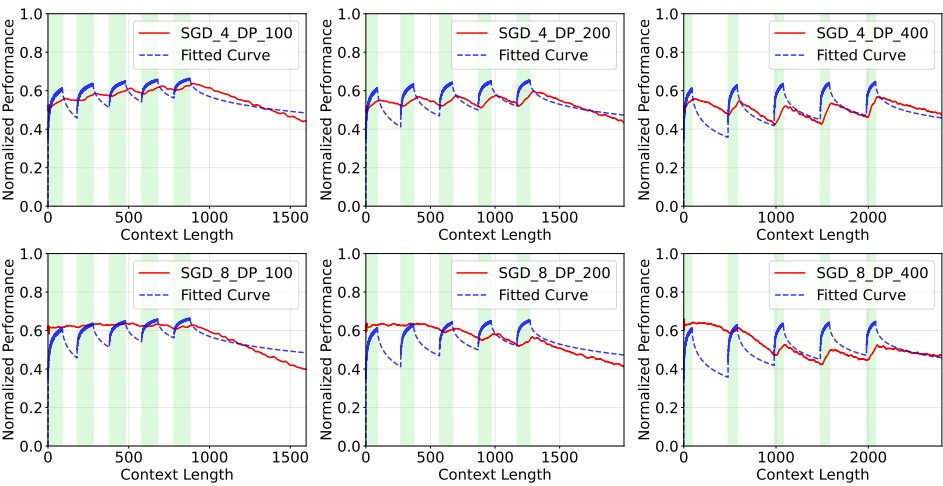

Figure 14: The performance comparisons between SGD and its fitted ACT-R model.

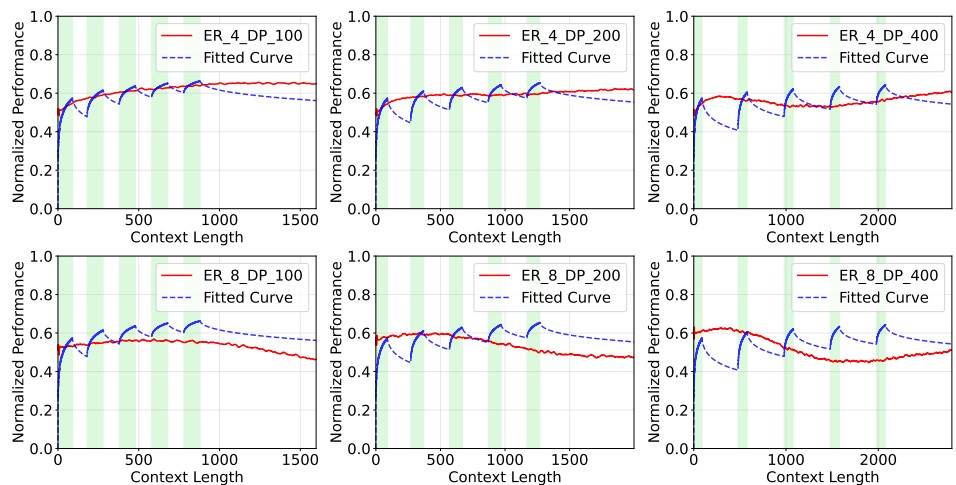

Figure 15: The performance comparisons between ER and its fitted ACT-R model.

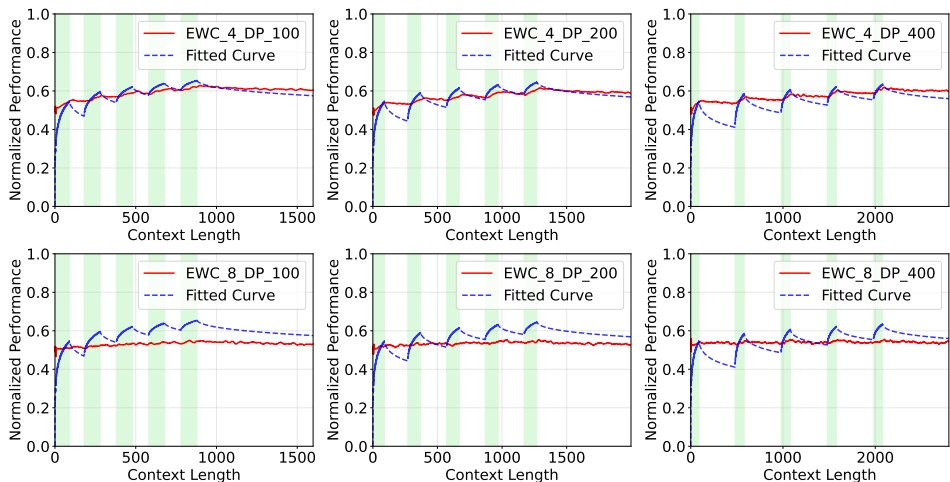

Figure 16: The performance comparisons between EWC and its fitted ACT-R model.

