# OpenReview forum: "In-Context Learning can Perform Continual Learning Like Humans"
_ICLR.cc/2026/Conference — Submitted to ICLR 2026_

### Official Review · Reviewer_3FtD · 2025-10-29

**Soundness:** 2
**Presentation:** 3
**Contribution:** 2
**Rating:** 4
**Confidence:** 3

**Summary:**

The paper proposes In-Context Continual Learning (ICCL), extending in-context learning (ICL) to handle sequential tasks without parameter updates. Drawing on human memory research, it studies how large language models retain information under different task schedules and finds that Distributed Practice yields a spacing “sweet spot” that enhances long-term retention, similar to human learning. A new Human Retention Similarity (HRS-MD) metric quantifies how closely model retention matches human memory patterns. Experiments on Markov-chain tasks show that while Transformer models achieve higher retention, linear-attention models (e.g., MAMBA, RWKV) exhibit more human-like forgetting curves. The study positions ICCL as a cognitively grounded, inference-only approach to continual learning.

**Strengths:**

1. This paper introduces an innovative concept, posing an interesting question: Can in-context learning itself support continuous learning without gradient updates? It successfully bridges cognitive science and artificial intelligence.
2. By introducing HRS-MD as a quantitative metric for “human-like similarity” in memory and linking ICCL with the interval effect and ACT-R memory model, this study establishes a robust, interpretable analogy between human memory retention and large language model behavior—a finding of significant value for understanding the emergent properties of ICL.

**Weaknesses:**

1. While this paper empirically observes knowledge retention patterns, it fails to fully elucidate the ICCL mechanism, such as how it supports sustained retention. Without theoretical analysis, it remains unclear whether ICCL reflects genuine information memory or merely short-term contextual reuse.
2. Based on the above, the lack of experimental comparisons with existing memory enhancement models makes it difficult to assess whether ICCL truly offers novel capabilities or merely replicates partial effects of these architectures under specific conditions.
3. The title and abstract assert that ICL “performs continual learning like humans.” However, the ICCL setup involves rearranging demonstrations within a single prompt rather than true knowledge transfer or consolidation over long timescales. This is conceptually closer to contextual rehearsal than to true continual learning, where the model updates its internal state across episodes. A more cautious framing might be advisable.

**Questions:**

1. Does ICCL's performance depend on the available context length? Specifically, how does its memory retention capability vary with context size? This is crucial for evaluating whether ICCL can achieve practical scalability beyond short sequence tasks. Additionally, how sensitive are the results in the paper to the selected hyperparameters?
2. Can the conclusions regarding “human-like continuous learning” be extended to real-world language or reasoning tasks?

---

> ### Author Response · Authors · 2025-11-20
> **Author Responses (Part 1)**
>
> **Weakness 1 (ICCL Mechanism Concern)**
>
> Thank you for raising this important point. Although ICCL does not rely on parameter updates, its retention behavior cannot be explained by short-term contextual reuse. As shown in Figure 2, once the LLM transitions from the target-task block to other tasks, its performance on the target task drops sharply—despite the fact that the target demonstrations were observed just moments earlier and remain relatively close to the query in absolute context position. This abrupt decline indicates that local proximity in the prompt or residual contextual traces are insufficient to account for ICCL’s retention patterns. In contrast, DP scheduling sustains retention by periodically reactivating the target-task representation throughout the sequence, effectively counteracting attention decay and cross-task interference. Moreover, the observed retention values exhibit a clear spacing “sweet spot” (Figure 5), and these retention dynamics closely match the human’s ACT-R model (Figures 10–13 in Appendix A.8). The emergence of these structured, parameterizable patterns and their alignment with well-established human retention curves provides strong evidence that ICCL reflects a genuine inference-time retention mechanism rather than a transient contextual artifact. This consistency with cognitive theory further supports that ICCL captures a meaningful and theoretically grounded form of in-context retention. We have added this clarification in Section 2.2 (Lanes 229-231) to make it clear.
>
>
> **Weakness 2 (Memory-Enhanced Model Comparison)**
>
> Thank you for the helpful suggestion. We agree that comparing against memory-enhanced architectures would be interesting, but such models fall outside the scope of the phenomenon we aim to study. The goal of our work is not to design or evaluate architectural memory modules, but to systematically analyze inference-time retention dynamics of standard LLMs without model-parameter updates or external buffers. The scheduling process in our experiments (SP, MP and DP) are all prompt-level scheduling strategies that do not modify LLM’s parameters or introduce any additional memory mechanism. Similarly, our gradient-based CL baselines (SGD, ER, EWC, PGP) operate through parameter updates within a small predictive model, rather than through architectural memory augmentation. In contrast, existing memory-enhancement models (e.g., retrieval-augmented systems and memory-augmented neural networks) explicitly introduce new external or architectural memory modules to extend a model’s inherent retention capabilities. We have added a clarifying discussion in Section 3.1 (Lanes 256-258) to explicitly differentiate ICCL from memory-enhanced models and outline why architectural comparisons are orthogonal to the scientific question addressed in this paper. We thank the reviewer for pointing out this distinction, which helped us improve the framing of our contribution.
>
> **Weakness 3 (Framing Overclaim Concern)**
>
> Thank you for highlighting this conceptual distinction. Our goal is not to assert that LLMs perform continual learning in the classical sense involving parameter consolidation, but rather to show that their inference-time retention dynamics under ICCL exhibit structured patterns analogous to human phenomena such as spacing effects and retrieval-based strengthening. ICCL operates entirely within a single prompt without weight updates, and its scheduling mechanisms allow an LLM to maintain long-term retention of a target task under cross-task interference. To prevent misunderstanding, we have revised the abstract (Lines 28–29) to explicitly clarify that ICCL focuses on long-term retention maintenance rather than parametric continual learning. We appreciate the reviewer’s feedback, which helped us refine the conceptual positioning of the work.

---

> > ### Author Response · Authors · 2025-11-20
> > **Author Responses (Part 2)**
> >
> > **Question 1 (Context & Hyperparameter Sensitivity)**
> >
> > Thank you for raising this important question. ICCL operates strictly within the context window of each evaluated LLM, and we have made this relationship explicit in the revised paper. In Section 3.1 (Lanes 287-294), we now clarify the rationale behind our parameter selections ($\varphi$ = 100, K = 5, $\varphi_I$ ∈ {10, 50, 100, 200, 400, 600}, $\varphi_D$ ∈ (0, 700)), which were chosen based on the token limits of different models (e.g., DeepSeek-R1 ≈ 4096 tokens, LLaMA3-8B ≈ 8192 tokens). This configuration ensures that all models can be evaluated within their effective context window and reach their optimal performance level, while still allowing controlled variation in $\varphi_I$ and $\varphi_D$ for cross-model comparison (Table 2 and Figure 5). Overall, ICCL maintains relatively high performance across different values of $\varphi_D$, and a clear sweet spot emerges when varying $\varphi_I$, which aligns well with empirical patterns observed in human memory research.
> >
> > To further examine the relationship between ICCL performance and context size, we conducted a new ablation study on the number of target-task blocks K. The results in Figure 3 reveal systematic and architecture-dependent trends: For Transformer-based LLMs (DeepSeek-R1, LLaMA3-8B), retention first improves and then declines as K increases, reflecting the effect of approaching their finite token limits. For linear-attention LLMs (RWKV-7, Mamba), retention increases more steadily and eventually stabilizes, consistent with these architectures’ greater efficiency in handling long sequences. Importantly, these trends are consistent across both simple and complex Markov-chain tasks. We have incorporated these clarifications and results into Section 3.2 (Lanes 374-377) to explicitly address context-length dependence and hyperparameter sensitivity.
> >
> > **Question 2 (Language/Reasoning Applicability)**
> >
> > Thank you for highlighting the importance of evaluating whether our conclusions extend beyond synthetic environments. To address this concern, we expanded our experiments by incorporating an additional benchmark built on the Regbench framework (Akyürek et al., 2024). Compared with markov-chain benchmark, Regbench introduces natural-language–style prompts with richer syntactic and structural variation, making it substantially closer to real-world sequential learning scenarios. As detailed in Appendix A.4 and A.6, we describe the prompt construction procedure and ICCL results on Regbench. The findings on Regbench show that ICCL continues to exhibit the similar qualitative behaviors (e.g., spacing benefits under DP and sharp decay under MP), indicating that the human-like retention phenomena observed in ICCL are not limited to synthetic tasks.
> >
> > We also clarify that our benchmark design intentionally departs from standard NLP tasks to align with the classical Ebbinghaus forgetting paradigm and to avoid reliance on pretrained parametric knowledge; typical one-shot or few-shot language tasks are too easy to serve as meaningful retention tests, whereas our chosen benchmarks offer a controlled setting for evaluating ICCL’s retention capabilities.

---

### Official Review · Reviewer_f1EV · 2025-10-30

**Soundness:** 3
**Presentation:** 3
**Contribution:** 3
**Rating:** 4
**Confidence:** 3

**Summary:**

This paper extends in-context learning to an inference-only continual learning setting, termed ICCL, through task scheduling and prompt rearrangement. Experiments on Markov-chain benchmarks show that ICCL benefits from distributed practice, revealing a human-like spacing effect that enhances retention. The proposed HRS-MD metric further quantifies similarity to human memory, showing that linear-attention models such as Mamba and RWKV exhibit more human-like retention dynamics.

**Strengths:**

The paper is well written and easy to follow. The study explores an interesting and novel perspective by connecting in-context learning with human memory theories. The idea of modeling retention dynamics through prompt scheduling and drawing parallels to human spaced learning is creative and provides valuable insights into the cognitive plausibility of large language models.

**Weaknesses:**

While the idea of connecting in-context learning with human memory is interesting, I’m not fully convinced that the proposed ICCL setup really fits the continual learning setting. Since the model doesn’t update its parameters and only relies on prompt scheduling or rearrangement, it feels more like a clever version of memory replay than real continual learning.

I have several concerns:

* From a CL standpoint, as illustrated in Fig. 1, when evaluating task i, the inclusion of demonstrations from other tasks seems unnecessary. I think the performance on task i should primarily depend on its own data, and adding unrelated task examples may introduce noise rather than benefit learning. If those experience of other tasks were removed, the setup would simply become standard ICL again — so it feels like the cross-task demonstrations are added mainly to make the setting look like CL. It’s not clear whether doing so is really necessary or meaningful.

* In the “massed practice” setup, I think simply reordering the experiences of the target task and other tasks might actually lead to better results. This way, the most relevant examples are placed closer to the query, which could even outperform the proposed distributed practice (DP) setting. Did you try this?

* Because there’s no training or parameter update, the model’s performance on new tasks depends entirely on a few in-context examples. This makes it hard to believe the method can generalize well to new or unseen tasks.

* The results could also be quite sensitive to how the “experiences” (both from target and other tasks) are selected. This is essentially the same issue as demonstration selection in ICL, and comparing to existing retrieval-based methods would make the evaluation more convincing.

Overall, the paper feels more like a cognitive-style study of LLM memory behavior than a real CL method. It’s interesting, but the framing as continual learning seems a bit forced.

**Questions:**

Please see the Weaknesses above.

---

> ### Author Response · Authors · 2025-11-20
> **Author Responses (Part 1)**
>
> **Weakness  (ICCL as Continual Learning?)**
>
> Thank you for raising this important question. ICCL aims to help LLMs to maintain long-term retention of a target task under cross-task interference without parameter updates. In long-context LLMs, forgetting arises at inference time: attention decay, recency bias, and cross-task interference can significantly weaken the influence of earlier target-task demonstrations, despite those demonstrations remaining in the prompt. Thus, retaining a high long-term retention on target-task across intervening tasks is a genuine and nontrivial challenge for LLMs. For this reason, ICCL is not merely replay. Pure replay would not change performance when the same demonstrations are simply reordered. In contrast, our experiments show that different scheduling patterns (SP, MP, DP) produce markedly different retention outcomes, because spacing and intervening tasks modulate the model’s inference dynamics and determine how strongly the target-task pattern is preserved. This demonstrates that ICCL captures inference-time forgetting effects that replay alone cannot account for. Therefore, ICCL represents a parameter-free yet meaningful form of continual learning that is unique to LLMs: it reveals whether the model can preserve long-term target-task competence under cross-task interference, a setting that standard independent ICL does not address. We have added this explanation to Section 1 (Lanes 64-69) to make the mechanism of ICCL convincing.
>
> **Weakness 1  (Necessity of Interference Tasks)**
>
> Thank you for the thoughtful question. The purpose of ICCL is not to maximize the immediate retention performance of “task i” but to maintain long-term retention of a target task under cross-task interference without parameter updates. In long-context LLMs, earlier demonstrations are not equally preserved—attention decay, recency bias, and cross-task interference can substantially weaken the influence of target-task examples even when they remain in the prompt. Therefore, the presence of other tasks is not superfluous noise but the very mechanism that creates the interference under which long-term retention can be meaningfully assessed. If all other tasks were removed, the setting would indeed reduce to standard independent ICL. Independent ICL has to repeatedly include the full target-task demonstrations to maintain long-term retention, producing prompts whose length grows linearly with the number of repetitions. ICCL, by contrast, achieves comparable retention with far fewer target-task demonstrations. This advantage becomes especially important when the model must maintain high retention over long content lengths but only a limited amount of target-task demonstrations is available (a common constraint in real-world few-shot scenarios). For these reasons, including other tasks in the prompt is both necessary and meaningful. We have incorporated this explanation into Section 1 (Lanes 64-69) and Section 2.1 (Lanes 199-203) to clarify the underlying motivation.
>
> **Weakness 2  (Reordering in MP Setup)**
>
> Thank you for the insightful question. In LLMs, positioning target-task examples close to the query is insufficient to maintain performance once intervening tasks appear. MP results in Figure 2 and Figure 6 clearly show that as soon as interference begins, retention drops sharply, even though the target block was recently observed and still lies relatively near the query in absolute context position. This indicates that the primary failure mode is not the lack of recency, but the model’s vulnerability to interference and attention decay over the continuation of the sequence. In contrast, DP distributes multiple smaller exposures of the target task throughout the sequence, producing repeated spaced reactivation rather than a single concentrated block. This spaced reactivation is precisely what prevents the abrupt performance collapse observed in MP. We have added this explanation to Section 3.2 (Lanes 322-323) for clarification.
>
>
> **Weakness 3  (Generalization Concern)**
>
> Thank you for the thoughtful comment. We clarify that ICCL is not designed to generalize to new or unseen tasks from a few in-context examples. Instead, the purpose of ICCL is to evaluate and enhance the model’s ability to retain high long-term retention of a target task while processing intervening tasks within a long context, thereby achieving continual-learning behavior without parameter updates. Long-context attention decay, recency bias, and interference from other tasks causes the model to lose the target-task pattern even though all parameters remain fixed. Our experiments show that such inference-time forgetting is substantial under SP or MP, and that DP mitigates this effect by repeatedly reactivating the target-task structure throughout the sequence. Generalization to unseen tasks is therefore outside the scope of ICCL and not required by the framework.

---

> > ### Author Response · Authors · 2025-11-20
> > **Author Responses (Part 2)**
> >
> > **Weakness 4 (Experience Selection Concern)**
> >
> > Thank you for the insightful comment. In our experiments, the “experiences’’ for both target task and interference tasks are not manually curated; instead, they are generated by sampling demonstrations from each task’s underlying state-generation mechanism (the probability transition matrix in the Markov-chain benchmark and the deterministic finite automaton in RegBench). For every task, we construct a task demonstration pool through this procedure and draw demonstrations independently for each run. To account for variability in experience selection, we repeat all experiments with 16 different random seeds and report mean performance together with confidence intervals. The consistency of results across seeds shows that our findings are robust to the choice of demonstrations, making the evaluation reliable even without retrieval-based augmentation. We have added Appendix A.4 (Lanes 838-1127) to explicitly describe the task-generation process and provide illustrative prompt examples for clarity. We also clarify that Experience Replay (ER), one of our baselines, is a retrieval-based rehearsal method within GBCL; despite using stored samples for retrieval, ER performs comparably to other GBCL methods and still exhibits long-range performance decay, further highlighting the distinct challenges addressed by ICCL.
> >
> > **Weakness (Continual Learning Framing)**
> >
> > Thank you for the valuable feedback. Our intention is not to position ICCL as a replacement for traditional gradient-based continual learning, but to study a complementary and LLM-specific form of continual learning that arises from inference-time forgetting in long contexts. While ICCL does not involve parameter updates, it captures a real continual-learning challenge for LLMs (maintaining a target-task mapping while processing many other tasks) and our experiments show that scheduling strategies such as DP significantly affect retention. We have revised the framing in the manuscript to make this clarification clear.

---

### Official Review · Reviewer_pMTu · 2025-10-31

**Soundness:** 3
**Presentation:** 2
**Contribution:** 3
**Rating:** 4
**Confidence:** 3

**Summary:**

This paper explores the capability of large language models (LLMs) to perform continual learning through in-context mechanisms, introducing the concept of in-context continual learning (ICCL). Drawing inspiration from human memory studies, the authors analyze how LLMs retain and accumulate knowledge across sequential multitask settings without parameter updates. Using Markov-Chain-based benchmarks, they demonstrate that ICCL exhibits human-like retention behaviors. They further introduce a human-retention similarity metric to measure alignment between model and human retention patterns, revealing that linear-attention models show more human-like retention despite lower absolute performance than Transformers. The findings suggest ICCL offers a cognitively grounded and practical, inference-only approach to continual learning that alleviates catastrophic forgetting.

**Strengths:**

The idea of in-context continual learning (ICCL) is novel and conceptually intriguing. The paper's design draws inspiration from cognitive psychology, adding an interesting interdisciplinary angle. Table 1 provides a clear and helpful comparison among gradient-based continual learning, standard in-context learning, and the proposed ICCL approach.

**Weaknesses:**

The motivation and design of ICCL need stronger justification. Essentially, ICCL remains a variant of in-context learning that includes demonstrations from multiple tasks rather than just the current one. While this may improve performance when the tasks are similar, it seems less effective when the tasks differ significantly. In such cases, using independent in-context learning for each task may yield better results, as it avoids the interference caused by unrelated demonstrations.

**Questions:**

1. At the beginning of Section 2.1, in the definition of $D_{\varphi}^{\tau}$, only $y$ carries the superscript $\tau$ (task index) while $x$ does not. Does this imply that different tasks share the same input distribution?
2. Section 2.3 is difficult to follow, particularly for readers without a background in cognitive psychology. A more accessible explanation or illustrative example would be helpful.
3. It is unclear why retention (or forgetting) should be a concern in ICCL. In gradient-based continual learning, parameter updates can cause forgetting, but in ICCL the parameters remain fixed. If one simply restores the demonstration samples for a previous task, its performance should, in principle, be fully recovered (implying zero forgetting). Please clarify this point.
4. The explanation of Eq. (6) is too brief. Since this equation is central to interpreting the experimental results, a more detailed discussion of its derivation and implications would strengthen the paper.

---

> ### Author Response · Authors · 2025-11-20
> **Author Responses (Part 1)**
>
> **Weakness (Motivation Justification)**
>
> We thank the reviewer for raising these insightful concerns. Below we clarify the motivation of ICCL, the role of task dissimilarity in DP scheduling, and the advantage of ICCL over independent per-task ICL.
>
> *The motivation of ICCL*: ICCL is motivated by a fundamental gap in our understanding of how LLMs behave when learning from long contextual streams. While ICL has demonstrated strong adaptability, prior studies largely examine single-task or one-off adaptation, and multi-task ICL work mainly analyzes how mixing or reordering demonstrations improves immediate prediction accuracy. These settings do not ask whether task knowledge can persist when new tasks intervene. GBCL methods explicitly address long-term retention but must contend with catastrophic forgetting. This gap motivates a fundamental question: does an analogous form of forgetting emerge purely during long-context inference, and can an LLM retain a target task when sequentially exposed to other tasks without any parameter modifications? ICCL is introduced precisely to study this overlooked problem. ICCL provides a framework to examine long-term retention under cross-task interference, offering a cognitively inspired view of how stable in-context knowledge can be maintained. This is essential for building general self-learning agents, which cannot rely on hand-crafted prompts for each task but must cope with continuously arriving, unpredictable tasks. We have revised Section 1 (Lanes 53-69) to more clearly articulate the motivation and design of ICCL.
>
> *Why DP uses dissimilar tasks*: DP in ICCL is not intended to teach the model the interference tasks. Instead, these tasks serve as spacing distractors, similar to human distributed practice. From cognitive theory (Pavlik & Anderson, 2005; Cepeda et al., 2008), spacing intervals are more effective when intervening materials provide clear contextual separation. Therefore, we select tasks with maximal dissimilarity in the benchmark to ensure more reliable reactivation of the target-task representation. We have added the above content in Section 2.1 (Lanes 195-198) describing this design choice and prompt examples of SP, MP and DP in Appendix A.4 (Lanes 836-954) for clarity.
>
> *Why ICCL outperforms independent ICL*: Independent ICL has to repeatedly include the full target-task demonstrations to maintain long-term retention, producing prompts whose length grows linearly with the number of repetitions. ICCL, by contrast, achieves comparable retention with far fewer target-task demonstrations, because DP naturally reinstates the target pattern through spaced exposure. This advantage becomes especially important when the model must maintain high retention over long content lengths but only a limited amount of target-task demonstrations is available (a common constraint in real-world few-shot scenarios). We have added the above explanations to Section 2.1 (Lanes 199-203) to make the advantages of ICCL over independent ICL explicit.
>
>
> **Question 1 (Unclear Variable Expression)**
>
> Thank you for the helpful comment. Our intention is not to assume that different tasks share the same input distribution; rather, each task may have its own distribution in our formulation. To avoid ambiguity, we have added the task superscript to variable “x” in the definition and revised the notation throughout the paper to ensure consistency.
>
>
> **Question 2 (Cognitive Psychology Background Gap)**
>
> Thank you for pointing this out. We now provide an intuitive, example-based explanation of the proposed HRS-MD in Appendix A.2 (Lanes 768-800), including an intuitive explanation of the ACT-R parameters (decay rate, activation noise, retrieval threshold) and a brief description of why Mahalanobis distance is used and how small/large HRS-MD values should be interpreted. We hope this makes Section 2.3 more accessible to readers without a cognitive psychology background.

---

> ### Author Response · Authors · 2025-11-20
> **Author Responses (Part 2)**
>
> **Question 3 (Inference-Time Forgetting Clarification)**
>
> Thank you for highlighting this important point. We clarify that the retention problem in ICCL arises from the inference process rather than parameter updates. Even though ICCL does not modify parameters, LLMs do not use all parts of a long prompt equally. Prior work has shown that LLMs exhibit strong position effects and long-context attention decay. As a result, earlier demonstrations become less influential as the context grows, even though they remain present in the prompt. Therefore, “restoring the demonstrations” does not guarantee full performance recovery: the model may attend to them weakly, or lose the task-specific pattern formed earlier in the sequence. Our experiments ( corresponding to SP and MP scheduling) confirm that target-task performance can drop significantly in long contexts unless the exposures are spaced appropriately. In short, ICCL studies a form of contextual forgetting (the gradual loss of the model’s ability to use earlier demonstrations during inference) and explores how to enable LLMs to maintain long-term retention under cross-task interference without parameter updates. We have added a clarification in Section 2.1 (Lanes 229-231) to make this distinction explicit.
>
>
> **Question 4 (Eq.(6) Explanation)**
>
>
> Thank you for pointing this out. We now clarify the motivation behind Eq. (6). Our goal is to measure retention in a way that is comparable across tasks with maximal dissimilarity. Bhattacharyya distance is used because it directly quantifies the overlap between the predicted and target distributions, but its raw magnitude varies widely across tasks. For this reason, we normalize the distance by anchoring it to two interpretable extremes: the best-achievable case (the model exactly matches the target distribution) and the worst-achievable case (the model collapses to a uniform distribution), representing complete forgetting. The exponential transform is introduced to linearize small differences in Bhattacharyya distance and to enhance sensitivity when distributions are close (an effect commonly used in similarity normalization). After normalization, the metric has a clear interpretation: 1 indicates perfect retention and 0 corresponds to uniform guessing. This expanded rationale has been added near Eq. (6) (Lanes 304-311). We hope this clarification makes the purpose and design of the metric fully transparent.

---

### Official Review · Reviewer_qTMj · 2025-11-01

**Soundness:** 3
**Presentation:** 3
**Contribution:** 2
**Rating:** 2
**Confidence:** 3

**Summary:**

This paper proposes In-Context Continual Learning (ICCL), an extension of ICL that lets LLMs retain knowledge across sequential tasks through task scheduling and prompt rearrangement, without updating parameters. Furthermore, a human-retention similarity metric (i.e., Human Retention Similarity via Mahalanobis Distance, HRS-MD) is proposed to evaluate alignment between human retention dynamics and continual learning methods.

**Strengths:**

- The retention characteristics of ICL in multi-task settings are extended to ICCL.
- The HRS-MD metric is proposed to quantify alignment with human memory dynamics.

**Weaknesses:**

- Overall, the ICCL method appears overly simple and lacks novelty, as the ordering of demonstrations has already been extensively studied (e.g., for multi-task settings [1] , others as noted in Section 4). Most importantly, the paper does not discuss the distinction between multi-task and single-task settings in the context of the demonstration arrangement problem. For instance, will the results differ if $\varphi_I$ is simply removed from DP? Furthermore, will the results be affected if different $\varphi_I$ (i.e., other tasks) are rearranged?
- To further validate the effectiveness of ICCL, more recent related GBCL baselines (e.g., EC-ACE [2], PGP [3], InsCL [4], etc.) should be discussed and compared. Some of these methods are applied to either Natural Language Processing or Computer Vision tasks, making them valuable for real-world applications.
- This paper only focuses on a single synthetic task (Discrete Markov Chains) and lacks real-world benchmarks, which significantly reduces the generalizability of the proposed method.
- The authors claim that, with larger $\varphi_D$, ICCL models exhibit more stable outcomes (see Lines 308–311). However, in Table 2, ER and EWC demonstrate greater stability compared to SGD, showing a marginal gap compared with ICCL methods. Does this suggest that the GBCL methods exhibit greater variability, and that incorporating more recent baselines (see Weakness above) could lead to different conclusions?
- Figure 3 is hard to understand, IMO. I was unable to distinguish between the results with and without task identifiers.
- The benchmark task of Discrete Markov Chains (DMC) should be discussed in detail. Additionally, on what basis are parameters (e.g., $\varphi, K, \varphi_D$) set? For instance, this paper does not include an ablation study on $K$.
Minor:
- Line 105: "a" $\rightarrow$ "an"
- Line 258: "$\varphi_I \in [...]$" $\rightarrow$ "$\varphi_I \in \{[...]\}$"
References:
[1] Xiong, Zheyang, et al. "Everything everywhere all at once: Llms can in-context learn multiple tasks in superposition." arXiv preprint arXiv:2410.05603 (2024).
[2] Caccia, Lucas, et al. "New insights on reducing abrupt representation change in online continual learning." arXiv preprint arXiv:2104.05025 (2021).
[3] Qiao, Jingyang, et al. "Prompt gradient projection for continual learning." The Twelfth International Conference on Learning Representations. 2024.
[4] Wang, Yifan, et al. "Inscl: A data-efficient continual learning paradigm for fine-tuning large language models with instructions." arXiv preprint arXiv:2403.11435 (2024).

**Questions:**

please see weakness.

---

> ### Author Response · Authors · 2025-11-20
> **Author Responses (Part 1)**
>
> **Weakness 1 (Novelty and Demonstration Effects)**
>
> We appreciate the reviewer’s constructive comments and have carefully clarified the conceptual distinction, experimental settings, and results accordingly.
>
> *Relative to Reference [1]*: Reference [1] studies how to mix or permute demonstrations to improve immediate in-context prediction. Our work focuses on long-term retention over sequential task exposure. In other words, this paper does not aim to optimize short-term accuracy but investigates whether an LLM can maintain a high long-term retention rate of a target task without parameter updates when required to shift away from its current other task. We have added Reference [1] in Section 4.1(Lanes 519-520) and the above clarification in Section 1 (Lanes 65-69) to make it clear.
>
> *Summary of novelty*: (1) ICCL provides, to our knowledge, the first framework that systematically investigates whether forgetting arises purely at inference time for LLMs and how to ensure them retain a target task under sequential cross-task interference without any parameter updates. (2) We introduce a human-retention similarity metric to quantify how closely a continual-learning process aligns with empirical human memory retention curves. We have added these details in Section 1 (Lanes 100-103) to make them clear.
>
> *Clarifying multi-task vs. single-task settings*: During the interleaved period (i.e., target-task learning period), removing other-task demonstrations ($\phi_I$ = 0) produces a single-task scenario containing only target task demonstrations (corresponding to MP setting), while keeping them ($\phi_I$ >0) forms a multi-task scenario containing both target task and other task demonstrations (corresponding to DP setting). Figure 2 illustrates that LLMs exhibit a sharp decline in target-task retention under cross-task interference, whereas the DP schedule substantially mitigates this effect and enables consistently higher long-term retention. We have clarified the above explanation in Figure 1 (Lanes 179-182) to illustrate how DP transitions to MP when $\phi_I$ is removed.
>
> *Effect of other-task duration*: We actually examined how varying the duration of other-task demonstrations from 10 to 600 affects retention and showed results in Figure 5. We can see that ICCL performance first increases and then decreases, revealing a “sweet-spot” phenomenon consistent with human spacing-effect theory.
>
>
> **Weakness 2 (Missing Modern Baselines)**
>
> We thank the reviewer for the suggestion to incorporate more recent GBCL baselines. We carefully evaluated their compatibility with our experimental setting. EC-ACE is specifically designed for vision-based continual learning and critically relies on augmentation-based multi-view inputs (e.g., random crops, color jitter, blur) to construct its consistency loss. Such augmentations cannot be directly applied to our discrete Markov-chain sequences without destroying the underlying transition structure. InsCL is designed for instruction-finetuning of LLMs using trainable adapters or LoRA modules, which falls outside the inference-only evaluation regime used in ICCL. In contrast, PGP provides a parameter-update–based GBCL method whose core mechanism can be directly implemented and evaluated as our existing baselines. Therefore, in the revised version, we include PGP as the modern GBCL baseline to complement SGD, ER, and EWC. Overall, PGP exhibits performance comparable to the other three GBCL methods and still suffers from long-range performance decay. We have added PGP’s results to Figure 2 and Table 2 (Lane 353 and Lane 361) to make the experimental comparison more complete. We also include a discussion of EC-ACE and InsCL in Section 4 (Lanes 507-509) to clarify their conceptual relationship to ICCL and to explain why they are not directly comparable within our setting.
>
>
> **Weakness 3 (Lack of Real-world Benchmark)**
>
> We thank the reviewer for emphasizing the need to evaluate beyond a single synthetic benchmark. In response, we have incorporated an additional evaluation using the RegBench framework (Akyürek et al., 2024), which introduces natural-language–style prompts and richer syntactic structure, providing a closer approximation to real-world sequential learning. Section 3.1 (Lanes 279-285) has been updated to highlight the complementary strengths of RegBench, and detailed prompt construction procedures and ICCL performance results are provided in Appendix A.4 (Lanes 836-1127) and A.6 (Lanes 1162-1187). Across this new benchmark, ICCL consistently exhibits similar qualitative retention patterns observed in DMC (such as spacing benefits under DP and sharp decay under MP). We also clarify that our benchmark design intentionally departs from standard NLP tasks to align with the classical Ebbinghaus forgetting paradigm and to avoid reliance on pretrained parametric knowledge; typical one-shot or few-shot language tasks are too easy to serve as meaningful retention tests.

---

> ### Author Response · Authors · 2025-11-20
> **Author Responses (Part 2)**
>
> **Weakness 4 (Variance Interpretation)**
>
> We thank the reviewer for raising this important point. To avoid potential confusion, we clarify that our analysis does not claim that ICCL exhibits lower numerical variance than ER or EWC. Rather, as shown in Table 2, ER and EWC quickly collapse into a low-retention regime and remain saturated there, which results in small fluctuations across different $\varphi_D$. In contrast, ICCL maintains substantially higher retention levels across all $\varphi_D$ settings, indicating stronger resistance to extended interference. To examine whether more recent GBCL baselines would alter this conclusion, we additionally evaluated PGP, one of modern GBCL methods compatible with our benchmark. PGP follows the same trend as SGD: it degrades sharply and remains in a lower-retention regime with increasing $\varphi_D$. We have revised our clarification accordingly in Section 3.2 (Lanes 367-372) to avoid any potential confusion.
>
> **Weakness 5 (Figure Results Misunderstood)**
>
> We appreciate the reviewer’s helpful comment. Figure 3 (now Figure 4) aims to compare the retention performance difference of ICCL models when prompts are provided with versus without task identifiers under SP, MP, and DP settings. Specifically, the plotted value represents the difference between ICCL’s retention performance with identifiers and that without identifiers. A positive value therefore indicates that the inclusion of task identifiers leads to higher retention performance. We have modified the figure caption in the revised manuscript (Lanes 403-404) to make the comparison more intuitive.
>
> **Weakness 6 (Lack of Benchmark and Parameter Setting Details)**
>
> We appreciate the reviewer’s helpful comments and have revised the paper accordingly.
>
> *Detailed description of the DMC benchmark*: We have updated DMC’s task descriptions in Section 3.1 (Lanes 272-275) and added contents about task generation mechanism and prompt examples of SP, MP and DP in Appendix A.4 (Lanes 838-954) to show how DMC benchmark works. The benchmark is designed to evaluate long-term retention in a controlled, interpretable sequence environment. Each task type corresponds to a state list with a specific transition matrix, and the model must predict the next state given the historical observed states. By controlling total state numbers in markov chains, the DMC benchmark allows us to test long-term retention under varying context lengths and task complexity levels.
>
> *Parameter setting rationale*: The parameter set in DMC benchmark is as below: K=5, $\varphi_I$∈{10,50,100,200,400,600}, $\varphi_D$∈(0,700). These values were determined by considering the token limits of LLMs (e.g., DEEPSEEK-R1 ≈ 4096 tokens, LLAMA3-8B ≈ 8192 tokens). This configuration ensures that all models can be evaluated within their effective context window and reach their optimal performance level, while still allowing controlled variation in $\varphi_I$ and $\varphi_D$ for cross-model comparison. We have added the rationale for choosing these parameter set in Section 3.1 (Lanes 287-294) to make parameter settings convincing.
>
> *New ablation study on K*: In response to the reviewer’s suggestion, we have conducted a new ablation study on K and included the results in Figure 3. Figure 3 presents retention performance of ICCL and GBCL methods ($\varphi_D$ = 600) for both simple and complex Markov-chain tasks, as the target-task block number K varies in {3, 5, 6, 7}. The results indicate that K has a substantially stronger influence on the retention behavior of LLMs than is typically observed in GBCL methods: For transformer-based LLMs (DEEPSEEK-R1, LLAMA3-8B), performance first increases and then decreases with larger K, since higher K implies longer contexts that can exceed the transformer’s token limit. For linear-attention LLMs (RWKV-7, MAMBA), performance steadily increases and stabilizes, as their architectures handle extended sequences more efficiently and has no token limit in theoretically. These patterns are consistent across both simple and complex DMC tasks. The analysis provides new insight into how target task demonstration numbers interacts with model architecture to determine retention behavior. we have added this ablation study results in Figure 3 (Lanes 378-390) and result descriptions in Section 3.2 (Lanes 406-412) to make this clear.
>
>
> **Weakness 7 (Typographical Errors)**
>
> Thank you for pointing out these minor issues. We have corrected the specific typos mentioned and carefully re-checked the entire manuscript to ensure that similar errors do not remain.

---

### Author Response · Authors · 2025-11-20
**Overall Clarification and Summary of Revisions**

We thank all reviewers for their insightful feedback. Across the revision, we have clarified the scope and positioning of our work: ICCL is not a new parametric continual-learning method, but the first framework aimed at understanding inference-time retention and maintaining long-term retention of a target task under sequential task interference. We strengthened the motivation, refined the framing in Sections 1–3, and added clearer descriptions of our benchmarks, task construction, and experimental setup in Appendix A2, A4 and A6. We also expanded analyses to show that the retention patterns we report (such as spacing effects and interference-driven decay) are robust across both synthetic and language-style benchmarks. These revisions address the key questions raised by reviewers and make the contribution and empirical evidence of ICCL more precise and transparent.

---

### Author Response · Authors · 2025-11-27
**Overview of Revisions with a Kind Request for Reviewer Consideration**

We thank all reviewers for their thoughtful feedback. In this revision, we have further clarified the positioning of ICCL as a framework for studying inference-time retention rather than a new parametric continual-learning method. We refined the motivation and framing in Sections 1–3, and added clearer descriptions of benchmarks and task construction in Appendix A2, A4, and A6. We also expanded analyses to show that the reported retention patterns remain consistent across both synthetic and language-style settings.

With the rebuttal deadline approaching, we kindly invite the AC and reviewers to take a look at these updates. We hope the revisions help clarify our contributions and address the questions raised.

---

### Meta-Review · Area_Chair_u6eY · 2026-01-05

**Summary:**

This paper has many weaknesses. I also took a look at the paper and agree with the reviewers and have some additional concerns.

**Reviewer Concerns:**

Reviewers broadly question the novelty, framing, and empirical strength of ICCL. They argue that ICCL largely reduces to in-context learning with reordered or mixed-task demonstrations, a setting already explored in prior work, and that its framing as continual learning is unconvincing since model parameters are never updated. Several reviewers view ICCL as closer to contextual rehearsal or memory replay than genuine continual learning. Empirically, the evaluation is considered weak, relying on a single synthetic benchmark, lacking real-world tasks, recent baselines, and sufficient ablations. Reviewers also question whether cross-task demonstrations are necessary or beneficial, especially when tasks differ, and suggest that simpler task-specific ICL or alternative ordering strategies may perform equally well or better.

I have also quickly read the paper and would like to raise the following additional concerns:

The paper does not clearly specify which continual learning setting it addresses (e.g., class-incremental, task-incremental, or domain-incremental learning).

Several in-context continual learning methods have been published in recent NLP venues, yet no comparison with these approaches is provided.

One reviewer noted the lack of comparisons with recent continual learning methods. While the authors argue that these methods operate under different settings or data, they nevertheless compare against early continual learning baselines such as ER and EWC, which raises questions about the consistency of the evaluation protocol.

**Reviewer Scores:**

I feel that the reviewers would not change their scores.

---

### Decision · Program_Chairs · 2026-01-26

Reject